# Current fossil fuel infrastructure does not yet commit us to 1.5 °C warming

Christopher J. Smith [1], Piers M. Forster [1], Myles Allen[2,3], Jan Fuglestvedt[4], Richard J. Millar[2,5], Joeri Rogelj [6,7,8] & Kirsten Zickfeld[9]

Committed warming describes how much future warming can be expected from historical emissions due to inertia in the climate system. It is usually defined in terms of the level of warming above the present for an abrupt halt of emissions. Owing to socioeconomic constraints, this situation is unlikely, so we focus on the committed warming from present-day fossil fuel assets. Here we show that if carbon-intensive infrastructure is phased out at the end of its design lifetime from the end of 2018, there is a 64% chance that peak global mean temperature rise remains below 1.5 °C. Delaying mitigation until 2030 considerably reduces the likelihood that 1.5 °C would be attainable even if the rate of fossil fuel retirement was accelerated. Although the challenges laid out by the Paris Agreement are daunting, we indicate 1.5 °C remains possible and is attainable with ambitious and immediate emission reduction across all sectors.

[1] Priestley International Centre for Climate, University of Leeds, Leeds LS2 9JT, UK. [2] Environmental Change Institute, Oxford University Centre for the Environment, South Parks Road, Oxford OX1 3QY, UK. [3] University of Oxford Department of Physics, Parks Road, Oxford OX1 3PU, UK. [4] CICERO, Postboks 1129 Blindern, 0318 Oslo, Norway. [5] College of Engineering, Mathematical and Physical Sciences, University of Exeter, North Park Road, Exeter EX4 4QF, UK. [6] International Institute for Applied Systems Analysis (IIASA), Laxenburg A-2361, Austria. [7] Grantham Institute for Climate Change and the Environment, Imperial College, London SW7 2AZ, UK. [8] Institute for Atmospheric and Climate Science, ETH Zurich, 8001 Zurich, Switzerland. [9] Simon Fraser University, Burnaby BC F5A 1S6, Canada. Correspondence and requests for materials should be addressed to C.J.S. (email: C.J.Smith1@leeds.ac.uk)

The Paris Agreement includes the aim to pursue efforts to limit global mean temperature rise to 1.5 °C above pre-industrial levels[1]. How close we are to 1.5 °C depends on the choice of reference (pre-industrial) period, the methods of generating global mean temperatures from historic records[2] and whether the human-induced warming contribution is isolated from the naturally forced warming and internal variability[3,4].

The zero emissions commitment[4–6] refers to the level of further warming that will occur if emissions abruptly cease, and is related to geophysical inertia. In previous studies, the long-term zero emissions commitment ranges from around −0.4 to +0.9 °C, and is sensitive to the carbon cycle response[7,8], ocean heat uptake[9], magnitude and pathway of historical warming[10], and whether or not non-$CO_2$ forcing is included[10,11]. When non-$CO_2$ forcing is included, setting emissions to zero leads to an initial rapid warming, associated with the removal of short-lived cooling aerosols, followed by a slower decline and stabilisation of temperatures driven by a reduction in the concentrations of short- and long-lived greenhouse gases[8].

An alternative assessment of committed warming is an infrastructure commitment[12–15]. This combines geophysical commitment concepts with knowledge of the possible speed at which fossil fuel-emitting infrastructure could be phased out. Under an infrastructure commitment, global society makes a serious effort to phase-out greenhouse gases but does not go as far as decommissioning power plants and other infrastructure before the end of their expected lifetimes (central estimate of 40 years for fossil fuel power plants)[12,13]. Transitioning to a zero carbon energy system within 40 years will be politically and societally challenging, and opinions are divided on whether this may be technically and economically possible[16–20]. We do not seek to assess the practical feasibility of this transition, but merely to report on the consequences in the context of keeping global mean temperature rise below 1.5 °C.

A third type of commitment that is often analysed is a constant forcing[21,22] or constant composition commitment, which is defined as the further warming that would result if atmospheric composition and hence radiative forcing were held fixed at today's values. Under such a scenario, temperatures continue to increase, with the rate of warming slowing down as equilibrium is approached[22]. The constant forcing commitment is not directly relevant to assessing warming committed from possible real-world mitigation scenarios, as constant forcing simulations require a continually declining but finite net greenhouse gas emission and would be hard to engineer. No known emission strategy gives constant forcing. Such constant forcing simulations have led to the misconception that inertia within the ocean system means that significant future warming is inevitable[6].

In this work, we explain that committed warming from present-day fossil fuel infrastructure is below 1.5 °C in 64% of an ensemble of scenarios with a simple climate model. These results are on the basis of fossil fuel assets starting to be retired from the end of 2018 once they reach the end of their design lifetimes, and accounts for the age distribution of extant stock. We focus on the energy generation, transport and industrial sectors which have the best data available for historical lifetimes and cover 85% of global emissions. The remaining 15% of global emissions are assumed to follow the retirement curves of the energy sector, as fewer data are available and in many cases there are cross-overs (for example, electricity supply and domestic heating). The phase out is such that the majority of $CO_2$ emissions have been eliminated in 40 years. As fossil fuel combustion emits short-lived climate forcers (SLCFs, which tend to result in a net negative forcing dominated by aerosol cooling) alongside $CO_2$, and both are gradually reduced in an infrastructure phase-out scenario, there is no sudden increase in warming from reducing emissions

gradually. This is in contrast to a zero emissions commitment, where the elimination of short-lived pollutants suddenly uncovers longer-lived greenhouse gas warming.

## Results

**Scenarios and model**. We define infrastructural commitments based on assumptions around today's carbon-emitting capital stock. In our central scenario, $CO_2$ emissions are phased out from the end of 2018 at a close to linear rate, becoming near zero after 40 years, with cumulative post 2018 $CO_2$ emissions of 195 GtC. In these scenarios, fossil fuel power plants, cars, aircraft, ships, and industrial infrastructure are replaced with zero carbon alternatives at the end of their life, and are phased out based on historical retirement data[13,23–26] and age profile distributions[13,27–29] (see Methods).

Scenarios are labelled in the form SSP < $n$ > - < year > - < commitment > (Table 1). The SSP label represents the Shared Socioeconomic Pathway scenarios[30], which are a set of narratives developed for integrated assessment models (IAMs) defining regional population growth, economic development and energy demand. From these narratives, IAMs produce a time series of global emissions over the 2010 to 2100 period. We investigate emissions under the SSP1 (green-growth[31]), SSP2 (middle-of-the-road[32]) and SSP3 (regional-rivalry[33]) narratives developed from the MESSAGE IAM[34]. The year label represents the start date of the commitment assumption, and is either 2018 or 2030, the latter investigating the effect of delayed mitigation action. For 2030 commitments, scenarios both with and without Nationally Determined Contributions (NDCs) for the 2020 to 2030 period are assessed[34]. NDCs are the set of submitted emissions reduction pledges by signatory countries to the Paris Agreement, and include unconditional NDCs (emissions reduction pledges made unilaterally) and conditional NDCs[35] (further emissions reduction pledges dependent on international cooperation and finance). Scenarios with conditional NDCs and unconditional NDCs are differentiated by appending a "c" or a "u", respectively, after the SSP scenario number.

For infrastructure commitments, we use the labels FAST, MID and SLOW to denote retirement rates for fossil fuel infrastructure. For the energy sector, these correspond to phase-outs over 30, 40 or 50 years based on the analysis of Davis and Socolow[13]. Different rates of fossil fuel phase-outs based on infrastructural lifetimes and age profiles for the transport and industrial sectors are used and also follow FAST, MID and SLOW trajectories (see Methods and Supplementary Figure 1), whereas a slower phase-out for agricultural emissions was assumed on the assumption that less mitigation potential exists for these predominantly non-$CO_2$ emissions[36]. In addition to FAST, MID, SLOW scenarios, a scenario with emissions set to zero instantaneously in either 2018 or 2030 (ZERO) is analysed for all SSPs considered. Furthermore, a CONST (constant forcing commitment) is analysed for SSP2-2018 and SSP2 with conditional NDCs starting in 2030 (SSP2c-2030).

This provides a total of 38 scenarios (3 SSPs × 3 NDC/start dates × 4 commitments, plus 2 CONST experiments for SSP2). The sector-dependent emissions of non-$CO_2$ greenhouse gases and SLCFs are also considered in our pathways (see Methods and Supplementary Figure 2).

Our phase-out scenarios are not bottom-up sectoral emissions projections or derived from detailed modelling. Instead, they assess the potential speed of emissions phase-out of each sector based on earlier literature and asset lifetimes. The scenarios assume mean rates of decarbonisation of 0.30, 0.23 and 0.18 GtC yr$^{-1}$ for the FAST, MID and SLOW pathways over the first 30, 40 or 50 years, respectively (Supplementary Table 1). In comparison,

| **Table 1 Summary of scenarios used in this study** | |
| --- | --- |
| **Shared socioeconomic pathway and start year of commitment** | **Commitment type** |
| SSP1-2018 | • ZERO: zero emissions commitment |
| SSP1c-2030 | • FAST: fast fossil fuel phase out (30 years for energy sector) |
| SSP1u-2030 | • MID: central fossil fuel phase out (40 years) |
| SSP2-2018 | • SLOW: slow fossil fuel phase out (50 years) |
| SSP2c-2030 | • CONST: constant forcing commitment (only for SSP2-2018 and SSP2c-2030) |
| SSP2u-2030 | |
| SSP3-2018 | |
| SSP3c-2030 | |
| SSP3u-2030 | |

the mean rate of emissions increase over 2006–2016 was 0.15 GtC yr$^{-1}$ (ref. [37]).

All scenarios are run using version 1.3.6 of the Finite Amplitude Impulse Response (FaIR) simple climate model[38,39]. FaIR includes a simple representation of biogeochemical feedbacks that reduce the efficiency of land and ocean carbon sinks with cumulative carbon uptake and increasing temperatures. The model produces $CO_2$ concentrations and effective radiative forcing (ERF) from its carbon cycle. ERF from non-$CO_2$ species are based on relationships between emitted greenhouse gases and SLCFs[39]. The generated ERF time series is compared with the ERF time series in Annex II of the Intergovernmental Panel on Climate Change (IPCC) Fifth Assessment Report (AR5), Working Group I[40] and a time-varying scale factor for forcing strength is applied throughout the historical period. This ensures that the best estimate anthropogenic non-$CO_2$ forcing matches the AR5 time series. This scaling factor decays linearly between 2000 and 2011, allowing a smooth transition from the AR5 forcing regime to the FaIR calculated values used for future projections. No historical scaling factor is applied to non-$CO_2$ greenhouse gases, as recent work[41] has shown that methane forcing is underestimated by ~25% in AR5 and we use the FaIR-generated ERFs throughout the historical and future periods.

To produce the pathways used in our analysis, emissions of greenhouse gases and aerosol and ozone precursors from the Representative Concentration Pathway (RCP) historical scenario[42] are used from 1765 to 2000, followed by emissions from the SSP1, SSP2 or SSP3 scenarios from 2005 to either 2018 or 2030. A linear transition between the historical and SSP emissions for each species between 2000 and 2005 is implemented. Solar and volcanic forcing are not included, as we model the anthropogenic part of climate change. Temperature anomalies since pre-industrial are calculated using a two time-constant representation[43–45] with a fast component representing the land, atmosphere and ocean mixed layer, and a slow component representing the deep ocean. The magnitude of temperature change due to the fast and slow thermal responses also depends on the equilibrium climate sensitivity (ECS) and transient climate response (TCR), which are input parameters to the model. Atmospheric greenhouse gas compositions and effective radiative forcing for the SSP2 zero emissions commitments and phase-out scenarios are shown in Supplementary Figure 3.

We use a 1000-member Monte Carlo sample of model input parameters for each scenario (Supplementary Table 2). These include the ECS, TCR, fast and slow ocean thermal response time constants, parameters governing the carbon cycle, and ERF strengths for 11 categories of anthropogenic forcing agents. The scaling of ERF of each forcing component ($CO_2$, $CH_4$, $N_2O$, minor greenhouse gases, tropospheric $O_3$, stratospheric $O_3$, aviation contrails, stratospheric water vapour, aerosols, black carbon on snow and land use) is varied based on their IPCC AR5

ERF uncertainty ranges[26]. To span carbon cycle uncertainty, ranges are defined that produce a plausible best estimate of concentrations of $CO_2$ in the present day of 407 ppm (ref. [39]), with a 5–95% range of 394 to 419 ppm. Although observations of $CO_2$ concentrations are more constrained than this range[46], allowing present-day $CO_2$ concentrations to assume a distribution of values lets us explore the uncertainty from the future carbon cycle response. This uncertainty leads to a range of ERF of ~10% of the best estimate, which is still only half of the ERF uncertainty for $CO_2$ quoted in AR5 (ref. [45]).

As working definition we employ a 1.1 °C human-induced warming estimate (the Global Warming Index (GWI))[3] by the middle of 2018 from a base period of 1850–1879 using the in-filled historical data set of Cowtan and Way[47]. We constrain the 1000-member ensemble by rejecting members that do not fall within the 5–95% uncertainty of the GWI of 0.94–1.32 °C over the 1850–1879 to 2018 period (Supplementary Figure 4). This results in 310 retained ensemble members for the SSP2 family of commitment scenarios (312 for SSP1 and 305 for SSP3, where differences relate to slight differences in the scenario evolutions between 2010 and 2018). Analysis of results is concentrated on the warming from 2018 until 2100.

**Committed 21st century warming**. Figure 1a compares the committed warming from 2018 from zero emissions (SSP2-2018-ZERO), constant forcing (SSP2-2018-CONST) and central infrastructural (SSP2-2018-MID) commitments under SSP2. Figure 1b shows the same commitments from 2030 (SSP2c-2030-ZERO, SSP2c-2030-CONST and SSP2c-2030-MID) when assuming SSP2 and the emissions currently implied the NDCs submitted under the Paris Agreement based on the central estimate of ref. [34].

In zero-emission scenarios, the warming effect of ocean inertia is found to be more or less cancelled out by removal of $CO_2$ from the atmosphere[48]. Such scenarios are useful in quantifying the geophysical commitment to past emissions but socioeconomic constraints render a zero emissions case extremely unlikely. There is a chance of considerable warming in the short-term owing to a sudden loss of aerosol-induced cooling, particularly if aerosol-induced ERF turns out to be at the more negative end of assessed uncertainty ranges[11]. The immediate climate response could vary from a slight cooling to a rapid warming, with a peak warming exceeding 0.4 °C above the present warming of 1.1 °C in 9% of ensemble members. By 2100, in contrast, 82% of ensemble members are cooler than at present, owing to a rapid decline in short-lived forcers. Excluding the possibility of strongly amplifying carbon cycle feedbacks such as permafrost melt or forest dieback, which are not included in our model, any peak temperature increase in such a scenario would be limited to 0.25 °C above present at the 75th percentile.

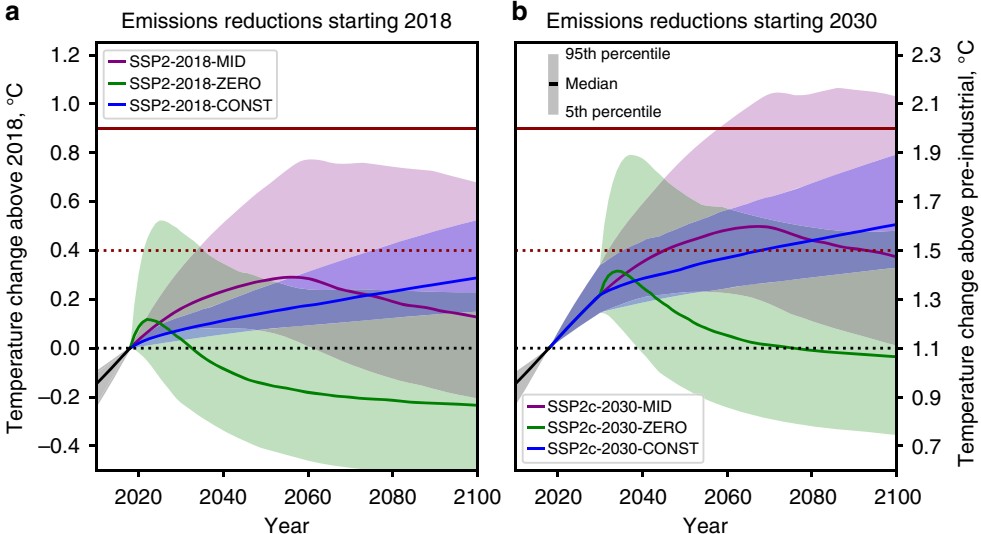

**Fig. 1** Three different commitment types from 2018 and from 2030. **a** Constant 2018 forcing commitment (SSP2-2018-CONST; blue), committed temperature change owing to default retirement of current fossil infrastructure (SSP2-2018-MID; purple) and zero emissions commitment (SSP2-2018-ZERO; green) all assuming the SSP2 baseline emissions pathway until 2018. **b** As **a**, but with commitments beginning in 2030 (SSP2c-2030-CONST, SSP2c-2030-MID and SSP2c-2030-ZERO), and Nationally Determined Contributions implemented from 2020 (based on ref. [34]). Shaded plumes show the 5–95 percentiles of the response under each scenario

Infrastructure commitment scenarios (Fig. 1, purple curves) do lead to a long-term warming commitment but they avoid the rapid increase in temperature caused by the sudden reduction in aerosol emissions, in the median case peaking when $CO_2$ emissions are close to zero[49]. In particular, SSP2-2018-MID, 64% of ensemble members show a peak temperature rise < 0.4 °C above 2018, although there is a large spread in the 5–95% range in both magnitude and timing of peak temperature change which is related to present-day aerosol cooling (Supplementary Figure 5). The constant forcing commitments (blue curves), although not realistic, show the often-misunderstood concept of committed warming in which temperatures continue to increase toward their eventual equilibrium after radiative forcing is stabilised[22]. In all scenarios, delaying action until 2030 results in an additional 0.25 °C of warming in the median case, and makes limiting peak temperature change to 1.5 °C more challenging.

In contrast to Matthews and Zickfeld[8] we find an end-of-century net cooling, rather than a net warming, for a zero emissions commitment, along with a smaller near-term and peak warming (Supplementary Figure 6 and associated commentary). This is owing to both the lower sensitivity of carbon sinks to temperature in FaIR than in the UVic2.9 model used in ref. [8], and the lower range of present-day aerosol forcing (5–95% range of −0.2 to −1.4 W m$^{-2}$; Supplementary Figure 5b) from our constrained projections, compared with the −1.0 to −1.5 W m$^{-2}$ range of ref. [8].

**Differing rates of fossil fuel retirement.** We investigate FAST, MID and SLOW phase-outs along with the ZERO commitment for SSP1 and SSP3 in addition. If the lifetime of fossil power plants is reduced by 10 years (FAST), resulting in post-2018 cumulative $CO_2$ emissions of 149 GtC, peak warming remains below 1.5 °C for all SSPs in > 67% of ensemble members (Fig. 2). Even if infrastructure lifetimes are extended by 10 years (SLOW; cumulative emissions 248 GtC), less than half of ensemble members exceed 1.5 °C. In addition, in the FAST phase-out scenarios, 2100 warming is below 1.5 °C in 83% or more of ensemble members in all SSPs and ~ 67% of ensemble members in the SLOW phase-out (Supplementary Figure 7).

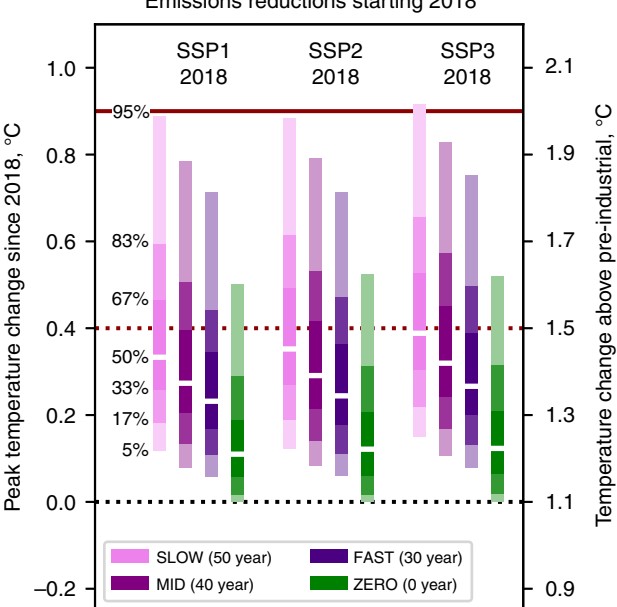

**Fig. 2** Peak temperature change compared with 2018 for different rates of fossil fuel phase-out (FAST, MID and SLOW) plus an abrupt cessation of all emissions (ZERO) from 2018. MID scenarios assume a phase-out of fossil fuel infrastructure based on historical generator lifetimes[13]. FAST and SLOW cases vary these lifetimes by subtracting and adding 10 years, respectively. Shown are results for emissions under SSP1, SSP2 and SSP3 until 2018. Supplementary Figure 7 shows the corresponding ranges for 2100 temperatures. SSP2-2018-MID and SSP2-2018-ZERO are consistent with the technological and zero emissions commitments in Fig. 1a

Delaying any infrastructural phase-out to 2030 results in more than half of ensemble members reaching a peak temperature exceeding 1.5 °C, even under a FAST phase-out (Fig. 3). However, under all of these scenarios, > 67% of the ensembles do not exceed

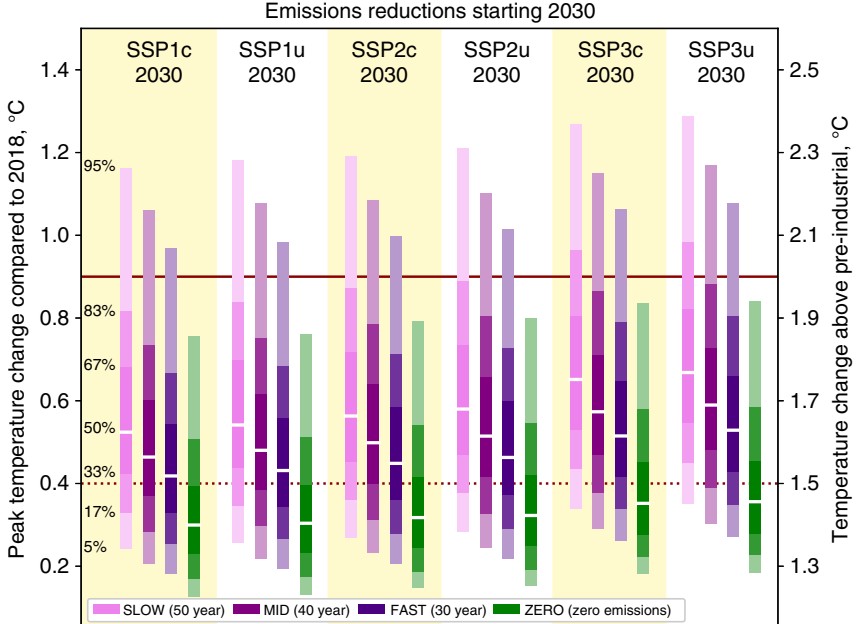

**Fig. 3** Peak temperature change compared with 2018 for different rates of fossil fuel phase-out (FAST, MID and SLOW) plus an abrupt cessation of all emissions (ZERO) from 2030. MID scenarios assume a phase-out of fossil fuel infrastructure based on historical generator lifetimes[13]. FAST and SLOW cases vary these lifetimes by subtracting and adding 10 years, respectively. Shown are results from SSP1, SSP2 and SSP3 with for conditional (c) and unconditional (u) NDCs based on socioeconomic developments under SSP1, SSP2 and SSP3. Supplementary Figure 8 shows the corresponding ranges for 2100 temperatures. SSP2c-2030-MID and SSP2c-2030-ZERO are consistent with infrastructure and zero emissions commitments in Fig. 1b

2 °C. As with the scenarios where phase-outs start in 2018, temperatures peak before the end of the century, and 2100 warming is < 2 °C in 83% or more of ensemble members except for SSP3 under a SLOW phase-out (Supplementary Figure 8).

None of these pathways explicitly rely on future negative emissions of $CO_2$. Nevertheless, carbon dioxide removal technologies may assist in either further speeding up the emissions reduction rates during the phase out, or offsetting $CO_2$ emissions in case infrastructure cannot be decommissioned sufficiently quickly.

**Model parameter uncertainty analysis.** The contributions to the warming commitments for peak temperature and temperature anomalies in 2050 and 2100 are analysed using a first-order variance-based sensitivity analysis (Fig. 4). We analyse the first-order sensitivities[50–53], which explain the contribution to the overall variance from the ECS/TCR, carbon dioxide forcing, aerosol forcing, other anthropogenic forcing, carbon cycle, and deep ocean thermal equilibrium time constant. In all cases, the sum of individual variance contributions does not sum to 100%, owing to interactions between parameters. In most cases ~ 75% of the variance can be explained by the first-order sensitivities.

For zero emissions commitments, the strength of present-day aerosol cooling is the dominant contributing factor in determining the magnitude of peak warming[8], describing between 40 and 60% of the variance in peak temperature projections. This is apparent in Fig. 1 and would suggest that uncertainties in future peak warming commitments can be better constrained if the present-day aerosol forcing is known with greater accuracy[11,54,55]. This finding is in contrast to high emission scenarios where the contribution of aerosol to the overall uncertainty in the temperature response becomes increasingly less important[56]. The prior distribution for year-2011 aerosol forcing is taken from the AR5 expert judgement assessed

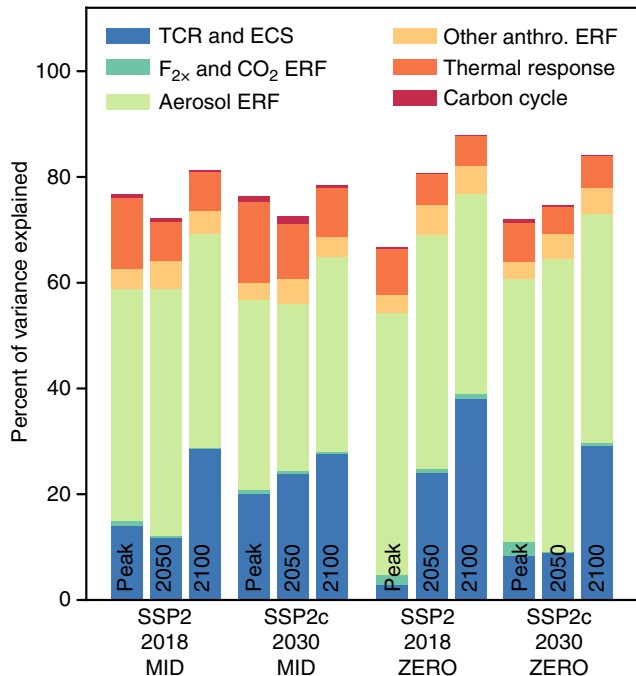

**Fig. 4** First-order variance-based sensitivity analysis of the peak, 2050 and 2100 temperature changes for infrastructure commitments and zero emissions commitments starting in 2018 and 2030. SSP2-driving scenario is assumed with NDC conditional commitments for 2030 start dates. Note that variances do not add up to 100%, as covariance terms account for the remainder

uncertainty range from a combination of CMIP5-era climate models and satellite observations. Climate models tend to produce estimates of aerosol ERF that are more negative than satellite observations, but methodological biases in both

procedures prevent a conclusive assessment of which product is more likely to be more representative of the true aerosol ERF[57]. If present-day (2018) aerosol forcing is towards the less-negative end of the uncertainty range, the zero emissions commitment in SSP2-2018-ZERO is small; a lower bound of $-1\,W\,m^{-2}$ produces a peak warming that remains under 1.5 °C across all ensemble members (Supplementary Figure 9). For some small negative values of present-day aerosol ERF, temperatures start to fall immediately after the zero emissions commitment suggesting that the decline in other SLCFs, mostly tropospheric ozone and methane, leads to a cooling that more than offsets a warming from aerosol removal under this scenario. For the SSP2-2018-MID infrastructure commitment there is a larger spread in peak temperature owing to the increased importance of other factors aside from the present-day aerosol forcing. These include climate sensitivity and the thermal response of the ocean, both of which have an extra few decades to act on the temperature evolution compared with SSP2-2018-ZERO. However, peak temperature remains below 1.5 °C in the SSP2-2018-MID ensemble when present-day aerosol ERF is less negative than $-0.6\,W\,m^{-2}$.

For end-of-century warming, the ECS and TCR also play an important role alongside the total present-day aerosol ERF, with contributions to total variance of between 30 and 40% for 2100 temperature change. Again, this is owing to the long-term action of climate sensitivity on temperature for a 2100 time horizon compared to a peak warming. ECS or TCR and aerosol ERF are anti-correlated in our constrained ensemble (Supplementary Figure 10), but both have proven difficult to constrain to a narrow range[58].

## Discussion

In this paper, we show that limiting warming to 1.5 °C is not yet geophysically impossible. Exceeding 1.5 °C occurs in only 9% of ensemble members under a zero emissions commitment if emissions cease at the end of 2018. Even if current fossil fuel infrastructure is retired at end of its lifetime and not replaced, it is possible to limit warming to 1.5 °C (bar the possibility of strongly amplifying carbon cycle feedbacks such as permafrost melt or forest dieback). Both reductions in $CO_2$ and other greenhouse gases are needed in order to take us close to such a temperature outcome. Aerosols exhibit a net cooling effect in the present day[11] but are co-emitted with greenhouse gases[59] and tropospheric ozone precursors. Although a gradual reduction of emissions from phasing-out fossil fuels does not result in a sudden warming, near-term (peak and 2050) warming commitments do depend strongly on the level of present-day aerosol forcing. End-of-century warming commitments depend on aerosol forcing, but with an important contribution from ECS and TCR.

The simple climate model used in our analysis may underestimate committed warming owing to possible future increases in climate sensitivity, as it employs a climate sensitivity that is invariant in time. Under a shifting pattern of surface warming through time[60], climate sensitivity over the historical period may have been lower than we can expect in the future as both the Eastern Pacific and Southern Ocean have not yet experienced strong warming[4,61,62]. Our simple model includes the biogeochemical feedbacks of decreasing carbon sink efficiency with increasing temperature and increasing biospheric and ocean carbon uptake[63], but may exclude possible other biogeochemical feedbacks where the elevated temperatures might affect future Earth system processes and carbon cycle response[64,65]. These feedbacks are expected to play out at timescales of decades to centuries and may thus be of limited relevance for estimating the committed warming over the 21st century. In this instance our results provide a useful first-order estimate of committed

warming based on current knowledge. Our findings contrast with a recent study by Pfeiffer et al.[15], which suggest that the infrastructure commitment from the energy sector alone is enough to commit us to warming above 1.5 °C. However, their analysis is based on allocating a share of the remaining carbon budget to 1.5 °C to energy sector emissions and does not discuss non-$CO_2$ emissions or pathways in non-energy sectors.

The challenge of making emission reductions across many sectors and countries remains very high but our committed warming scenarios show two important insights. First, geophysics does not yet commit the world to a long-term warming of $>1.5$ °C. Second, even when phasing out existing $CO_2$-emitting infrastructure at the end of its expected lifetime, warming is also kept to below 1.5 °C (or 0.4 °C warmer than today) with $>50\%$ probability, whereas delaying action until 2030 reduces this probability to below 50%. This is important information, as it shows that whether global mean temperature increase will be kept to below 1.5 °C depends on societal choices made today and emissions reductions implemented over the coming decades.

## Methods

**Fossil fuel phase-out scenarios.** The ZERO and CONST scenarios are trivially set to zero emissions or constant forcing in the year of commitment and warrant no further explanation, so we detail construction of the FAST, MID and SLOW phase-out scenarios below.

**$CO_2$ emissions.** In each phase-out scenario starting in 2018, $CO_2$ emissions from fossil fuels decline at an approximately linear rate of 0.30 (FAST), 0.23 (MID) or 0.18 (SLOW) GtC yr$^{-1}$ from their 2018 peak (Supplementary Figure 1), consistent with energy sector fossil fuel phase outs over 30, 40 or 50 years respectively. These rates are scaled up for the 2030 phase-out scenarios to achieve phase-outs over the same time periods. The near-linear rate of decline in $CO_2$ emissions is partly coincidental as each sector is treated differently as detailed below. We extend the analysis of ref. [12], which considered emissions impacts of fossil fuel energy generation. At 42% of global $CO_2$ emissions in 2015 (ref. [66]), energy generation is the largest sector but by no means the only significant one. We separately consider transport (24% of 2015 emissions) and industry (19% of 2015 emissions), covering 85% of global $CO_2$ emissions. The remaining 15% of emissions (from residential, service and other aggregated sources) are combined with the energy sector in our scenarios. This is on the basis that it is less straightforward to find robust lifetime assumptions from these sectors, and in the absence of better information we proceed on the assumption that any transformation that could be implemented in the energy sector could also be put into effect elsewhere at the same rate.

Land use-related $CO_2$ is abruptly set to zero in the first year following the phase-out start. This is on the basis that humanity would be concerned enough under a phase-out scenario to prevent additional deforestation. In the context of the simple climate model used in this study this is a conservative estimate, as land use change (a negative forcing) scales with cumulative land use $CO_2$ emissions[39].

**Energy generation.** The FAST, MID and SLOW rates of fossil fuel energy phase-out are taken from Davis and Socolow[13] based on the historical precedent of power plant retirements over 30, 40 and 50 years. The MID profile represents their central 40-year estimate and is the one which we give most weight to in this study. Retirement profile curves from these scenarios were scaled in order to achieve the 30, 40 or 50 year phase-out from the 2018 or 2030 baseline emissions, which differ slightly to the absolute values in Davis and Socolow who conduct their analysis from 2012. Our scaled-up emissions project an energy sector commitment of 94 GtC in 2018 based on a 40-year power plant retirement age.

Pfeiffer et al[15]. recently conducted a study of committed emissions in the energy generation sector using more up-to-date knowledge, concluding there was 82 GtC of committed emissions in power plants currently in operation and a further 74 GtC of emissions either planned or under construction. We only consider emissions from currently operating plants in this study. Our figure of 94 GtC is therefore a slight overestimate of Pfeiffer et al.[15], which is explained by a slower than expected growth in installed capacity in the last few years.

**Transport.** The shares of total transport $CO_2$ emissions due to road transport (75%), aviation (11%) and shipping (11%) in 2015 is taken from ref. [66] and used as our starting point. The 3% of global transport emissions that do not fall under one of these three classes is grouped with road transport.

**Road transport.** We assume that the majority of road transport emissions are due to passenger cars. Globally, the number of cars manufactured has grown at an exponential rate of 3% per year, based on data from 2000 to 2017 (ref. [25]). We

assume a historical scrappage rate based on passenger car data from the US which continues into the future. This is a modified logistic curve:[28]

$$F(t) = \frac{1}{L + B \exp(-\kappa t)} \qquad (1)$$

where $t$ is age in years and $F(t)$ defines the probability that a car of age $t$ is scrapped that year. We make the simplified assumption that every car on the road contributes equally to $CO_2$ emissions, and that in the present day the number of alternative fuel vehicles is a negligible proportion of the world fleet. By applying the assumptions for year-on-year growth of number of vehicles produced with the rate of historical scrappage, we obtain an estimate of the age profile for the current fleet. We freeze this in the first year of our phase-out scenarios and impose that no new fossil fuel powered cars are manufactured. The current fleet is then retired at the rate of eq. (1) given their 2018 (or 2030) age profiles, resulting in a yearly decline in the number of petrol and diesel cars on the road and associated $CO_2$ emissions.

In the MID case, which we take to be the best estimate of the parameters in ref. [28]. ($L = 2.724$, $B = 314.03$, $\kappa = -0.275$), this results in a mean vehicle lifetime of 15.6 years. We also vary the coefficients in eq. (1) for the FAST ($L = 4.654$, $B = 440.658$) and SLOW ($L = 0.794$, $B = 187.402$) phase outs based on $2\sigma$ values of these parameters in ref. [28] ($\kappa$ is unmodified to maintain a logistic-shaped curve). These produce retirement curves with mean vehicle lifetimes of 12.7 and 18.0 years, respectively.

**Aviation**. A similar method is applied to aircraft where we assume a 2018 or 2030 age profile based on historical growth rate and scrappage. We assume an exponential growth rate in the number of aircraft of 4.2% per year, which has been applicable for the last 30 years[24,27]. The retirement profile for aircraft again follows a logistic curve[27], which defines a mean aircraft lifetime of 26 years for our MID phase-out. There is no assumed retirement of aircraft in the first 5 years of operating life in ref. [27]. To define our FAST and SLOW phase-outs we shift the retirement profile by ± 5 years.

**Shipping**. An yearly exponential growth rate of 3.7% in shipping tonnage is estimated based on data[29] between 2000 and 2016. No data on the retirement curve of ships are available, but a review of asset lifetimes in several countries[26] suggests ships are typically in service for 25–30 years. We therefore use the same retirement profiles as aircraft for the FAST, MID and SLOW phase-outs of 21, 26 and 31 years, respectively.

**Industry**. Estimates of the existing lifetime of industrial infrastructure are not abundant, and we are therefore limited to using one estimate from the expected lifetimes of cement kilns of between 30 and 50 years[23]. The limitations of this are acknowledged, as industrial emissions cover a wide range of sub-sectors (manufacturing, metal processing, paper, chemicals, to name a few), with different profiles of emissions species. Reducing emissions to net zero in processes requiring heat input would be challenging without carbon capture and storage, on which our scenarios seek to avoid any explicit dependence.

We take an approach similar to transport lifetimes for industrial infrastructure based on eq. (1), with mean retirement ages of 30, 40 or 50 years for FAST, SLOW and MID and standard deviation of 6 years in all cases.

As the mean of the logistic function represents the year in which half of industrial infrastructure is retired, industrial emissions reach net zero at a slower rate than energy emissions, which are phased out entirely after 30–50 years and transport emissions which follow logistic curves with shorter mean lifetimes (Supplementary Figure 1). This is a more conservative assumption, which reflects the increased difficulty of estimating the technical feasibility of phase-outs from this sector.

**Non-CO$_2$ emissions**. We model the change in $CH_4$, $N_2O$ and SLCFs by using the fraction of each emissions species from each sector in 2008 in the Emissions Database for Global Atmospheric Research (EDGAR) v4.2 database[67,68] (Supplementary Table 3). SLCFs act as tropospheric ozone and aerosol precursors and include $SO_2$, CO, non-methane volatile organics, nitrous oxides, black carbon, organic carbon, and $NH_3$. For the phase-out scenarios, the change in non-$CO_2$ emissions is shown in Supplementary Figure 2.

**Energy, industry and transport**. For energy generation and industry sectors, the sectoral fractions in Supplementary Table 3 are applied to the total emissions of each species in the SSP scenario. In phase-out scenarios, emissions are scaled by the ratio of $CO_2$ emissions from the energy or industrial sectors, respectively, to the $CO_2$ emissions from the first year of the phase-out. This treatment therefore assumes that non-$CO_2$ emissions are co-emitted with $CO_2$ in the same ratios as in 2018 or 2030.

A similar treatment is provided for road transport and non-road transport, in which phase-out of road and (aviation plus shipping) non-$CO_2$ emissions are treated individually and scaled to their corresponding $CO_2$ emissions phase-out.

**Agriculture**. The infrastructure commitment in the agricultural sector is the hardest to estimate. On the one hand, agricultural practices, diets and their associated emissions could in theory be ceased over the course of a couple of seasons or years. On the other hand, the necessity to continue food production to sustain our global population and the low amount of mitigation options that are currently identified for this group of greenhouse gas emissions suggest that a significant amount of agricultural emissions might persist throughout the remainder of the century[69]. Here we take a middle-of-the-road approach to estimate the infrastructure commitment of the agricultural sector. Livestock emissions could in theory be reduced quickly, by slaughtering all meat animals. Taking the lifespan of meat cattle as 36 months, we assume a linear slaughter rate, therefore giving a phase out of livestock emissions over 3 years. Non-livestock related agricultural emissions result primarily from fertiliser usage for $N_2O$ and rice production for $CH_4$[70]. Emissions from agriculture are linearly phased out over 82 years, so that they reach zero in 2100 under the 2018 phase-out scenarios. For consistency, this rate is not altered for the phase-out scenarios beginning in 2030, so agricultural emissions reach zero in 2112.

The marginal cost abatement curves for emissions reductions from agriculture show limited opportunity to make deep emissions cuts from existing technologies[71,72]. We here do not model this explicitly. Bringing agricultural emissions down to zero would, however, have to rely on a combination of changing diets, technological improvement and overall reduction in global population. For sensitivity cases, we assume alternative pathways: emissions consistent with RCP2.6, a constant 2018 level of non-livestock emissions, and agricultural emissions set immediately to zero (both with and without a 3-year phase out of livestock emissions). Varying these assumptions results in a variation in 2100 temperature change of between −0.08 and + 0.12 °C different to our default assumption (Supplementary Figure 11). We do not change our default 82-year phase out between the FAST, MID and SLOW scenarios.

**Biomass**. Consistent with setting land use related $CO_2$ to zero immediately, we assume that no more land is deforested, and biomass-related emissions of other species are also abruptly zeroed.

**Other sectors**. Consistent with our use for $CO_2$ emissions, we scale all sectors not elsewhere considered with energy emissions.

**Climate model**. We use the FaIR model (version 1.3.6)[38,39] to evaluate all future scenarios. FaIR has been validated against the behaviour of more complex carbon cycle and earth system models[38,73] and has been designed to emulate the historical effective radiative forcing relationships from the IPCC Fifth Assessment Report[40,45] given input emissions. FaIR produces similar 21st century temperature projections to the more established Model for the Assessment of Greenhouse gas Induced Climate Change (MAGICC6)[42] for the Representative Concentration Pathway scenarios[42] as shown in ref. [39], with the agreement particularly good for the lower-end RCP2.6 and RCP4.5 scenarios, which are most relevant to this analysis.

FaIR uses a simplified four time-constant representation of atmospheric $CO_2$ concentrations based on the impulse response model used in Chapter eight of the IPCC AR5 Working Group I[74]. The atmospheric lifetime of $CO_2$ in FaIR increases with increasing temperature and cumulative carbon emissions, reproducing the behaviour seen in contemporary Earth system models[75]. ERF from non-$CO_2$ greenhouse gases[41,45], tropospheric ozone[76], stratospheric ozone[42], stratospheric water vapour oxidation from methane[45], aviation contrails[77], aerosols[78,79], black carbon on snow[80] and land use change[39] are calculated from simple relationships or models based on annual, global totals of input emissions of 39 greenhouse gases and SLCFs.

The emissions time series are used to calculate greenhouse gas concentrations and ERF. For 1765–2000 we scale the best estimate ERF time series generated by FaIR to match the extended AR5 time series exactly, which corrects for small variations in the efficiencies of natural carbon sinks[81] and changes in spatial patterns of aerosol forcing[82].

From the ERF, global mean surface temperature change is calculated. Temperature anomalies at each timestep $t$ are composed of slow (deep ocean; $d_1$) and fast (upper ocean, atmosphere and land; $d_2$) contributions to the temperature change

$$T_{t,i} = T_{t-1,i} \exp\left(\frac{1}{d_i}\right) + q_i F\left(1 - \exp\left(\frac{1}{d_i}\right)\right); i = 1, 2 \qquad (2)$$

$$T_t = T_{t,1} + T_{t,2} \qquad (3)$$

where $F$ is efficacy[83]-weighed total ERF, and the $q_i$ coefficients are the contributions to temperature change from the fast and slow components, which

depend on ECS, TCR and ERF from a doubling of $CO_2$ $F_{2\times}$ (ref. [84]):

$$q_1 = \frac{1}{(k_1 - k_2)F_{2\times}}(\text{TCR} - k_2\text{ECS}) \qquad (4)$$

$$q_2 = \frac{1}{(k_1 - k_2)F_{2\times}}(k_1\text{ECS} - \text{TCR}) \qquad (5)$$

$$k_i = 1 - \frac{d_i}{69.66}\left(1 - \exp\left(-\frac{69.66}{d_i}\right)\right); i = 1, 2 \qquad (6)$$

We use an efficacy of 1 for all forcing components except black carbon on snow for which we use an efficacy of 3 (ref. [80]), and 69.66 years is the time to a doubling of $CO_2$ under a compound 1% per year increase in $CO_2$ concentrations, consistent with the definition of TCR.

The CONST (constant forcing) commitments are performed by running the model to 2018 or 2030, saving the ERF and contributions to fast and slow temperature anomalies output in that year, and re-running the model from 2018 or 2030 with just the forcing to temperature routines, bypassing the emissions and carbon cycle to ERF calculations. This means that in the CONST commitment there is no feedback from increasing temperatures on the carbon cycle past the date of commitment, consistent with the definition of constant forcing.

**Ensemble generation.** Uncertainty in ensemble projections is driven by the uncertainty in the input parameters to FaIR. ECS and TCR are drawn from a joint lognormal distribution[85] (correlation coefficient 0.81) informed by the ECS and TCR from the abrupt4xCO2 (instantaneous quadrupling of $CO_2$ concentrations) and 1pctCO2 (compound annual 1% increase in $CO_2$ concentrations) results from CMIP5 climate models[39,86,87]. The slow and fast time constants of ocean thermal response ($d_1$ and $d_2$) are drawn from normal distributions based on the analysis of ref. [44], and carbon cycle response parameters drawn from normal distributions based on ref. [38]. The scaling factors for ERF uncertainty for 11 different anthropogenic forcing components are drawn from normal, composite normal or lognormal distributions and are informed by AR5 estimates[45]. In total, 1000 sample parameter sets are drawn and the model spun up for the historical period and then projected forward. Ensemble members not falling within the historical uncertainty of observational temperature change[47] are rejected.

The posterior (temperature-constrained) distributions of each input parameter are shown in Supplementary Figure 5, with the correlations between parameters and each parameter with 2100 temperature change in Supplementary Figure 10. In the experiment design, uncertainties are uncorrelated with each other except for ECS and TCR. In reality there is a weak positive correlation between $d_1$ and ECS (or TCR), but constructing a joint distribution with one lognormal variable and one normal is problematic, so we take $d_1$ to be uncorrelated with ECS. We find there is a negative correlation in the posterior distributions between ECS or TCR and present-day aerosol ERF, which is expected[39,84,88], and positive correlations between total ERF and both aerosol ERF and $F_{2x}$, although as these factors are not independent this is also expected.

**Global mean surface temperature change.** Following ref. [3] we use the Global Warming Index with observations from Cowtan & Way to estimate an anthropogenic contribution to temperature change of 1.076 °C from 1850 to 1879 average in May 2017. The Global Warming Index removes an estimate of the natural component of forcing from the temperature record, and as such our baseline is slightly higher than observational datasets including the unmodified Cowtan & Way estimate. At current rates, the anthropogenic warming should have reached 1.1 °C sometime in mid-to-late 2018, and we use this figure as our 2018 level. We find that future temperature projections in the Representative Concentration Pathway scenarios are insensitive to which observational data set is used for constraint[39], but our results do depend on our starting point and how close we currently are to 1.5 °C, which in turn depends on how global mean surface temperature is defined[2].

Cowtan & Way use observations from HadCRUT4 (ref. [89]), comprising of sea surface temperatures over the ocean and near-surface air temperatures over land, in-filled for missing data (blended, in the notation of ref. [2]). The missing data is from regions of the world where no observations exist. The polar regions, where observations are sparse, are warming faster than the rest of the planet. Therefore, the HadCRUT4 observations (blended-masked, in the notation of ref. [2]) tend to produce lower estimates of the observed warming than blended (and unmasked) datasets such as Cowtan & Way. Providing that global temperature observational coverage continues to increase, and that the in-filling method of Cowtan & Way is sufficiently accurate, the blended-masked observations will converge towards the blended observations in the future.

A third possibility is using near-surface air temperatures globally, over ocean regions as well as the land (tas-only in the notation of ref. [2]). This reflects the estimates of global mean surface temperature usually reported from climate models, and is typically higher than blended and blended-masked observations because the air above the sea surface warms faster than the ocean surface itself. A

drawback of this method is that there are limited historical observations of near-surface air temperature over the ocean, and any observational data set would have to be infilled and likely correlated with data from other sources such as climate models.

**Variance-based sensitivity analysis.** To determine the contributions to overall variance in Fig. 4, we use the 310 retained members of the SSP2 scenario to obtain the lower and upper bounds of each of the 18 parameters in the FaIR model that produce at least one ensemble member within the observed warming trend (Supplementary Table 4).

We perform a first-order Sobol' sensitivity analysis[50] using inputs of each of the 18 varying parameters from a Saltelli sampling scheme[51–53]. For each experiment we generate 2500 samples of each variable, requiring 50,000 model runs in total. The sampling process, although informed by constrained values of the parameters, may produce combinations of parameters that are inconsistent with observed warming, but we do not constrain these sensitivity runs by historical temperature in order to fully investigate the response of the model. The sensitivity analysis is performed for both the peak temperature change and the temperature change in years 2050 and 2100.

**Code availability.** The FaIR model is available from [https://github.com/OMS-NetZero/FAIR]. FaIR version 1.3.6 is used for all simulations in this paper. The analysis code used to produce results in this paper, and all supplementary data described above, is available from [http://doi.org/10.5281/zenodo.1565230].

## Data availability
The RCP historical emissions time series are available at http://www.pik-potsdam.de/~mmalte/rcps/. SSP emissions datasets can be obtained from the Integrated Assessment Modelling Consortium (IAMC) Scenario Explorer at https://data.ene.iiasa.ac.at/iamc-1.5c-explorer/. EDGAR emissions data (v4.2) can be obtained from http://edgar.jrc.ec.europa.eu/overview.php?v=42. Temperature observations from the Cowtan & Way data set were downloaded from http://www-users.york.ac.uk/~kdc3/papers/coverage2013/had4_krig_annual_v2_0_0.txt.

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

## Acknowledgements

We acknowledge financial support from the Natural Environment Research Council under grant NE/N006038/1.

## Author contributions

CJS and RJM built the model, CJS ran the simulations and wrote the manuscript. PMF designed the study. All authors contributed substantially to the text, discussion and interpretation of results.

## Additional information

**Competing interests:** The authors declare no competing interests.

