## [Peer Review File · Nature Communications]

Reviewer #1 (Remarks to the Author):

Review of '1.5C – are we there yet?' by Smith and co-authors.

In this contribution the authors use a simple carbon-cycle model to investigate various forms of global warming commitment with a focus on the probability of exceeding 1.5 degrees above pre-industrial. In contrast to previous studies (e.g. Matthews and Zickfeld 2012), it is found that committed warming in the case of an abrupt emissions halt is more likely than not less than the present-day temperatures. This happens in the applied model, on average, despite the rapid warming from diminished aerosol cooling, and it is surprising because previous modelling studies that found nearly constant temperature in centuries following ceased emissions even without the aerosol warming effect.

Thus, the study must present a case as to why this unexpected model result is, or is not, likely to be relevant to the real world?

The only observational constraint placed on the used models is global warming until present. But there are other observables that could help constrain the model ensemble, e.g. atmospheric CO₂ concentrations, planetary radiation imbalance, or perhaps carbon uptake by the oceans. As I understand it, also other atmospheric constituents are calculated based on their emissions and so the evolution of their atmospheric concentrations. I would further, out of interest, also like to see the posterior distributions of TCR, ECS, aerosol forcing and timescales of temperature adjustments. My point is that there could easily be internal compensations between e.g. excessive carbon uptake and small aerosol cooling in the historical setting that leads to observed warming, but smaller 2100 commitment.

There further seems to be something fundamentally wrong with the implementation of the constant composition commitment case. If atmospheric composition is held fixed at, say, 2018 levels then it is not to be expected that there is an abrupt increase in temperatures. Instead warming should continue but at a slowing pace as the deep oceans equilibrate.

Given the title contains the word 'yet', I didn't quite understand how the infrastructural commitment is the most relevant case. Surely, most governments (perhaps except the US) around the world can shut down fossil fuel emissions rather quickly, either indirectly by introducing high carbon taxes or directly by forbidding the activity, as it is done with most other pollutants once identified as such. But that is besides the point, the word 'yet' to me implies 'up until now'. If to be retained, though, the large paragraph on page 3 must be made more readable.

Overall, I found it difficult to sort out what was new, what was assumed and how I could interpret these model results. I would suggest adding analysis as outlined above and re-writing the manuscript

with more emphasis on why this model does what it does, rather than the current presentation which leaves the reader in the cold concerning interpretation. This would be a fairly major effort, but seems conceivable.

Minor comments:

- Before submitting this again the authors need to add line numbers to the document.
- All figures are of very low quality (blurry), consider using less compression, or a vector format.

Reviewer #2 (Remarks to the Author):

Review of 1.5 °C – are we there yet? by C. J. Smith et al

Summary:

A simple carbon-climate model is used in a large ensemble (with varied model parameters) to explore three future carbon emission scenarios in relation to the 1.5 °C maximum warming target set by the Paris Agreement to avoid the most dangerous consequences of anthropogenic warming.

The manuscript finds that we have not yet reached the point at which 1.5 °C becomes impossible. The authors consider both scenarios where fossil fuels are immediately ceased and more realistic scenarios where fossil fuels are phased out as the existing infrastructure is decommissioned and replaced by non-fossil fuel emitting alternatives.

Other recent studies have considered what happens to warming when carbon emissions are suddenly and completely ceased. However, the consideration here of more realistic scenarios makes the findings unique and important, and definitely relevant to the multi-disciplinary audience of Nature Communications.

There are some substantive issues that I would like further clarification on before supporting publication of this manuscript. I feel that clarification on these issues will help readers in the climate modelling field place the findings of this manuscript in context, while also providing additional confidence on the findings for inter-disciplinary readers.

Therefore, my recommendation is to return to the authors for clarification of the issues raised below.

Substantive points:

Point (1): The interrelationships between the TCR, ECS and ocean ventilation timescales:

The authors define a TCR and ECS for each model simulation within their ensemble. I understand that the authors link the TCR in some way to the ECS (i.e. they are co-varied), based on the statistical link in the CMIP5 ensemble.

However, the timescales of ocean tracer uptake are also important in setting how the TCR and ECS relate. For example, a system with fast ocean ventilation will have smaller TCR than a system with slower ocean ventilation if the ECS is held constant – since in a system with infinitely fast whole-ocean equilibration for heat and carbon the TCR and the ECS would be the same thing.

This means that the TCR should be a function of the ECS (which it is in their ensemble) and a function of the two ocean ventilation timescales (which it is not in their ensemble).

Can the authors justify why they should vary the ocean ventilation timescales independently from TCR/ECS relationship? Surely the TCR should emerge from the ocean ventilation timescales and the ECS?

Or, more likely for their simple model set up, surely the ocean tracer uptake timescales should be co-varied depending on the difference between the TCR and the ECS.

(i.e. In your ensemble design, the difference between the TCR and ECS should be greatest when the ocean ventilation timescales are slowest, and the difference between the TCR and ECS should be smallest when the ocean ventilation timescales are quickest).

Point (2): Using a model with defined TCR to investigate situations in which CO₂ is not increasing at 1% per year.

The Transient Climate Response (TCR) is defined as the warming relative to a preindustrial steady state when atmospheric CO₂ has reached double preindustrial levels, via a pathway of continuously increasing atmospheric CO₂ at an exponential rate (at 1% per year every year).

The simple carbon-climate model (FAIR) used by the authors requires the value of the TCR to be defined. The model then calculates warming over time based, in part, on this defined value of TCR.

However, the authors use their simple carbon climate model to investigate a system that is very different from the continuous exponential increase in atmospheric CO₂.

The whole concept of TCR is undefined (and is therefore not applicable) for a system in which emissions suddenly reduce/cease and atmospheric CO₂ remains steady or falls.

It is unclear (so far as I know) how the emerging value of the TCR in complex climate models relates to the time evolution of surface temperatures in a scenario after emissions are suddenly ceased.

Can the authors provide some clarification as to how the TCR is affecting the evolution of surface temperatures in their simple carbon-climate model simulations after emissions cease? (Presumably this is in part defined by the equations used by the model?).

And can they then discuss whether influence of the TCR on warming in a situation in which the TCR is undefined is reasonable?

At the very least the authors should present an additional figure in the supplementary material, similar to figure S1, showing the variation in committed warming with the model values of TCR and ECS.

It would also be good to see a supplementary table that showed the correlation of different model input parameters to the committed warming (e.g. the R² correlation coefficient for each model input property versus committed warming).

This would help the reader to assess what is leading to the variation in future warming in the model ensembles, and fully understand the results of this study.

Point (3): Relationship to existing literature

There are existing studies in the literature that look at how warming continues (or not) after emissions are suddenly ceased, e.g.:

Ehlert, D., & Zickfeld, K. (2017). What determines the warming commitment after cessation of CO₂ emissions? *Environmental Research Letters*, 12, 015002.

Frölicher, T. L., Winton, M., & Sarmiento, J. L. (2014). Continued global warming after CO₂ emissions stoppage. *Nature Climate Change*, 4, 40–44.

Williams, R. G., Roussenov, V., Frölicher, T. L., & Goodwin, P. (2017). Drivers of continued surface warming after cessation of carbon emissions. *Geophysical Research Letters*, 44.

The results of this study ought to be compared to this existing literature. Does the simple model ensemble used here behave consistently with the complex model simulations considered in Frölicher et al (2014), or the complex model simulation and simple model ensemble considered in Williams et al (2017)?

If the simple model used here behaves consistently with other studies when emissions are suddenly ceased, then this would provide additional confidence for the findings of this study in the unique and important new scenarios where emissions are phased out as infrastructure is decommissioned.

Point (4): Future projections from this model ensemble for the standard scenarios

What are the future projections for this simple model ensemble for a standard RCP scenario (e.g. RCP8.5 or RCP2.6), and how do these compare with existing model projections?

It would be good to see how this particular ensemble does, compared to existing results, when projecting warming for a standard future scenario. This would again provide additional confidence in the results of this study, and provide more context for how climate scientists should interpret the results presented here with existing results in the literature.

Note that I do not think that disagreement with existing projections for a standard scenario would mean that the results of this study were less important. Just that, for other climate scientists to interpret the results of this study in context, we should see how this model ensemble does for scenarios we are familiar with.

This context is especially important for interpreting the probabilistic language of the manuscript (see minor point below). Therefore, could a supplementary figure be shown where the simple model ensemble projections for a standard RCP scenario are compared to existing projections from other model ensembles in the literature.

Point (5): Atmospheric CO₂ at equilibrium

The simple model used here applies an impulse response function to determine atmospheric CO₂ over time in response to carbon emissions. The way this is described in this study seems to suggest (to me) that the final atmospheric fraction of CO₂ is fixed by the impulse response function, and is the same for any size of carbon emissions.

However, due to the non-linearity of ocean carbonate chemistry, the final atmospheric fraction of CO₂ varies with emission size.

(As the cumulative emission size grows the final atmospheric fraction of emitted CO₂ also grows, until emissions reach around 5000PgC. This means that the final atmospheric fraction for a cumulative emission size of 500PgC is less than the final atmospheric fraction for a cumulative emission size of 5000PgC, e.g. see Goodwin, P., R. G. Williams, M. J. Follows, and S. Dutkiewicz (2007), Ocean-atmosphere partitioning of anthropogenic carbon dioxide on centennial timescales, *Global Biogeochem. Cycles*, 21, GB1014, doi:10.1029/2006GB002810).

If my reading of how the model used here works is correct, can the authors reveal what the final atmospheric fraction of CO₂ is from their impulse response function, and whether this is consistent with a cumulative emission size similar to the size implied in their scenarios (something like 600 to 700PgC)?

I may of course have read this wrong, and the impulse response function used by the model may account for the non-linearity of carbonate chemistry. In which case, can the authors just clarify how this is done in the methods/supplementary section.

Minor point:

Point (1): Writing style:

I find the writing of the manuscript in places lacks clarity and context. For example, consider the following statement from the second paragraph of the Main text (beginning “Figure 1 compares...”):

“Figure 1 compares the committed warming from 2018 (panel a) with the committed warming from 2030 when assuming the emissions currently implied by the Nationally Determined Contributions (NDCs) submitted under the Paris Agreement (panel b,...)”

Figure panels 1a and 1b do not show this. What figure 1 panel (a) and (b) do show is projections of committed warming from a particular model, configured in a particular way to perform ensemble experiments, under these two scenarios. This context is important, and should certainly be included within this paragraph – preferably before referencing Figure 1. Something like “We use a simple climate model (Ref) to perform ensemble experiments (Methods) and project future committed warming for different scenarios. Figure 1 shows our projections of committed warming for ...”.

Elsewhere, there are references to “X% chance” and “X% probability” of warming outcomes for a particular scenario. Again, this language is used without context, and reads as though the authors have constructed definitive probabilistic warming projections. What the authors mean is that “X% per cent of simulations”, or “the X percentile of simulations”, in their ensemble have a particular warming outcome under that scenario.

This is an important distinction, and one that readers should be made aware of explicitly within the main text of the article, without having to read the methods/supplementary material to assess whether the authors have actually shown definitive probabilistic outcomes.

Reviewer #3 (Remarks to the Author):

Smith et al. tackle the salient question of whether committed warming is in conflict with the international target of avoiding a 1.5degC in mean temperatures. Although I am sympathetic to their arguments, I have serious misgivings about the manuscript that I think should prevent it from being published without major revisions.

My overarching criticism is that the authors' main conclusion—that the 1.5deg target “still remains geophysically possible with ambitious emission reduction”—is not well supported by their methods. This could in principle be remedied by either much clearer descriptions of methodological limitations or by using more rigorous methods, but I suspect the former approach would take a lot of punch out of the manuscript (perhaps so much that Nature Communications would not be interested in

publishing it), and the latter would entail substantial further work which the authors might not want to do.

A secondary criticism is that the text itself is also not very well written, with confusing and thin explanations of the methods and data, and discussion that is frequently garbled by ambiguous definitions and too-brief methodological descriptions.

The main methodological concerns I have are (1) the climate model and (2) the estimation of committed emissions.

Although the author team includes some very well-respected climate modelers, there is very little validation of this (unpublished) model in the manuscript. The result that most concerns me is that—judging by Fig.1—global mean temperatures begin to decrease within 40 years of a cessation of emissions. Unless these runs include large net negative emissions (not clear), this seems contrary to previously published results of at least some full GCMs and EMICs that show approximately stable mean temperatures for several centuries after net-zero emissions are achieved (see, e.g., Hadley and UVic results in Matthews and Weaver, NatGeo, 2010). I could certainly be persuaded that the model is correct and accurately reflects the latest and greatest CMIP models, but the dearth of validation combined with the importance and contradictory nature of this result raise red flags.

At least as important is what amounts to a concerning mischaracterization of “phase-out” scenarios in the text. Throughout the main text, the authors present these scenarios so as to suggest an update of the bottom-up analysis of global fossil fuel infrastructure in Davis et al. 2010., e.g. discussing power plant lifetimes. However, the supplementary materials suggest that what they have actually done is to apply the various rates of emissions reductions in the previously published analyses to current fossil emissions. This method, however, will drastically underestimate committed emissions because the age distribution of fossil fuel burning infrastructure has changed substantially since 2010—e.g., many new power plants have been built, particularly in China and the U.S. (see Davis and Socolow, 2014) which have decreased the rate at which emissions will decline when a given lifetime is assumed.

The climate model results seems to represent a low-end estimate of future warming and the committed emissions are calculated in such a way that they underestimate future emissions. Neither of these possible biases are conservative to the authors’ very strong claims that 1.5deg is geophysically possible with ambitious mitigation efforts. For this reason, as I stated earlier, I do not support publication of the manuscript as written and recommend substantial revisions either to the claims or the methods (or at least a clear and robust justification of the current methods).

Minor comments:

- I think it'd be easier to read Fig. 1 if the various commitments were separated into other panels.
- use "natural" and "technical" lifetime for infrastructure. I don't think either of these is quite right. "Historical" or perhaps "design" or "expected" lifetime would all be preferable. But at least be consistent.
- discussion of SSPs and "NDC-conditional trajectories" are especially opaque and could do with a sentence or two more explanation.

References:

- H. D. Matthews, A. J. Weaver, Committed climate warming. *Nature Geoscience* 3, 142-143 (2010).
- S. J. Davis, K. Caldeira, H. D. Matthews, Future CO₂ Emissions and Climate Change from Existing Energy Infrastructure. *Science* 329, 1330-1333 (2010).
- S. J. Davis, R. H. Socolow, Commitment accounting of CO₂ emissions. *Environmental Research Letters*, (2014).

Reviewer #1 (Remarks to the Author):

Review of '1.5C – are we there yet?' by Smith and co-authors.

In this contribution the authors use a simple carbon-cycle model to investigate various forms of global warming commitment with a focus on the probability of exceeding 1.5 degrees above pre-industrial. In contrast to previous studies (e.g. Matthews and Zickfeld 2012), it is found that committed warming in the case of an abrupt emissions halt is more likely than not less than the present-day temperatures. This happens in the applied model, on average, despite the rapid warming from diminished aerosol cooling, and it is surprising because previous modelling studies that found nearly constant temperature in centuries following ceased emissions even without the aerosol warming effect.

Firstly, we thank the reviewer for their time spend in their very useful and thorough review of our manuscript.

The two main reasons for the differences in our zero emissions commitment (ZEC) to that in Matthews and Zickfeld (2012) is in the strength of the aerosol forcing and sensitivity of the UVic2.9 carbon cycle to temperature change. In figure 3 of Matthews and Zickfeld (2012), the aerosol forcing is varied from -0.8 to -1.9 W m^{-2} . In their model setup, a -0.9 W m^{-2} present-day aerosol forcing results in a temperature change of about zero in 2100, and -0.8 W m^{-2} results in a net cooling. These levels of aerosol forcing produce historical time series that are too cool in Matthews and Zickfeld (2012); their aerosol forcing is more negative to balance other forcings or possibly a high ECS. Matthews and Zickfeld (2012) also acknowledge that values above and below their constrained range are possible because of uncertainties in the climate and carbon cycle response; furthermore they used a single EMIC, UVic2.9, which has a single value of the TCR (1.9 K), whereas we use an ensemble of values that span the AR5 plausible range of TCR/ECS and also sample the response in the carbon cycle.

To return to the aerosol discussion, -0.9 W m^{-2} is our year-2011 best estimate of aerosol forcing, as we follow AR5, and by 2018 when the zero emissions commitment starts it is closer to -0.8 W m^{-2} . Following Matthews and Zickfeld (2012), it would not be unexpected that our zero emissions commitment best estimate is negative for a -0.8 to -0.9 W m^{-2} aerosol forcing.

Following also comments from the second reviewer, we have added in a short review of zero emissions commitments in the introduction.

In the supplementary information we add supplementary figure 6 with associated commentary, repeating some of the analysis of Matthews and Zickfeld (2012) with the FAIR model.

Thus, the study must present a case as to why this unexpected model result is, or is not, likely to be relevant to the real world?

Arguably, ZECs are not relevant in the real world but we include ZECs to demonstrate a geophysical commitment. In context of the argument above we argue that these results are not unexpected, but the results of a different aerosol forcing and carbon cycle model than Matthews and Zickfeld (2012).

The only observational constraint placed on the used models is global warming until present. But there are other observables that could help constrain the model ensemble, e.g. atmospheric CO₂ concentrations, planetary radiation imbalance, or perhaps carbon uptake by the oceans. As I understand it, also other atmospheric constituents are calculated based on their emissions and so the evolution of their atmospheric concentrations. I would further, out of interest, also like to see the posterior distributions of TCR, ECS, aerosol forcing and timescales of temperature adjustments. My point is that there could easily be internal compensations between e.g. excessive carbon uptake and small aerosol cooling in the historical setting that leads to observed warming, but smaller 2100 commitment.

The FAIR model (version 1.1) is described in Smith et al. (2017). This paper is undergoing a second review which describes version 1.2 and it is possible that it will be accepted in Geoscientific Model Development in the near future. In this response we comment with reference to the discussion paper (v1.1).

The carbon cycle parameters in FAIR are selected such that the atmospheric CO₂ concentrations between pre-industrial and present day are representative of observations. This is shown in figure 4a of Smith et al. (2017). When spun up with historical emissions (as it is for all simulations in this paper), the model calculates CO₂ concentrations of 407 ppm (2.5 to 97.5% range of 394 to 421 ppm) for 2017. The median is in line with observations (from e.g. NOAA), whereas the range of simulated emissions is much wider than the observational uncertainty. It should be borne in mind that the variation in CO₂ concentrations is the mechanism for sampling the uncertainty in CO₂ forcing, and assuming the logarithmic relationship of CO₂ concentrations to forcing, the range of ± 13 or 14ppm around the best estimate accounts for less than a 10% difference in forcing, in line with the uncertainty in radiative forcing from CO₂ (and half of the 20% uncertainty assumed for effective radiative forcing). This argument is now added to the main manuscript.

Figure 10 in Smith et al. (2017) shows the correspondence with the earth's energy imbalance compared with best estimates from CERES (for the top of atmosphere) and Argo (for ocean) for FAIR, and that it is well-constrained over the 2005-2015 period.

FAIR does not track the level of carbon uptake by ocean. The airborne fraction (total emissions minus amount taken up by land and ocean) however can be calculated, and is shown in figure A below for the SSP2-2018-MID phase-out scenario as defined in the revised version. Between 1960 and 2000, the airborne fraction is around 0.45, which is in good agreement with chapter 6 of AR5. The rise in airborne fraction between 2000 and 2015 is a symptom of a switch from the RCP historical to SSP2 scenario, the latter of which has higher CO₂ emissions in the early 2000s than the RCPs, and the sudden drop in 2018 a result of setting land use CO₂ emissions abruptly to zero.

Figure A | Airborne fraction of CO₂ from SSP2-2018-MID

Following the reviewers' suggestion we have included posterior distributions of ECS, TCR, F_{2x} , total ERF, aerosol ERF and the deep ocean thermal equilibrium time constant (d_2) as a supplementary figure S5. We have shown the correlation coefficients between each parameter, and between each parameter and 2100 temperature change, in figure S9. Most parameter combinations are poorly correlated except for where this is part of the ensemble design (e.g. correlation between ECS and TCR). This shows that any internal compensation is limited.

There further seems to be something fundamentally wrong with the implementation of the constant composition commitment case. If atmospheric composition is held fixed at, say, 2018 levels then it is not to be expected that there is an abrupt increase in temperatures. Instead warming should continue but at a slowing pace as the deep oceans equilibrate.

Following the reviewer's comment, we have investigated this in more detail and agree that it was indeed not correct. The updated figure 1 shows the correct future profiles for a constant composition case (implemented as a constant forcing commitment) with the global mean surface temperature gradually approaching equilibrium.

Given the title contains the word 'yet', I didn't quite understand how the infrastructural commitment is the most relevant case. Surely, most governments (perhaps except the

US) around the world can shut down fossil fuel emissions rather quickly, either indirectly by introducing high carbon taxes or directly by forbidding the activity, as it is done with most other pollutants once identified as such. But that is besides the point, the word 'yet' to me implies 'up until now'. If to be retained, though, the large paragraph on page 3 must be made more readable.

We believe that it would be highly unlikely that governments would shut down fossil fuel generation quickly due to a combination of market forces, technology lock-in, vested interests and (lack of) political willpower. The technology lock-in component is isolated in this study with the fossil fuel retirement scenarios, although we make no assumptions about the economy or consumer behaviour (or whether a zero-carbon world is technically possible in the time scales we investigate, or at all, but Jacobson and Delucchi (2011) suggest this is possible).

Understanding the reviewer's confusion, we have changed the title to "Committed warming and the 1.5°C Paris Agreement target" to emphasise that the paper is about committed warming but that we are not measuring whether we have reached this limit so far.

The large paragraph in question has been split up and re-written along with much of the main manuscript, so it is hoped that this now appears clearer.

Overall, I found it difficult to sort out what was new, what was assumed and how I could interpret these model results. I would suggest adding analysis as outlined above and re-writing the manuscript with more emphasis on why this model does what it does, rather than the current presentation which leaves the reader in the cold concerning interpretation. This would be a fairly major effort, but seems conceivable.

Thank you for your valuable suggestions. We include a more extensive description of the model in the Methods, and point the reviewer to Smith et al. (2017) for a full model description. The model code is available from GitHub at <https://github.com/OMS-NetZero/FAIR>. In context of applying the model to these scenarios, we include additional diagnostics of posterior distributions from the model output (fig S5), correlations between parameters (fig S9), outputs of atmospheric GHG concentrations and effective radiative forcing (fig S3), and temperature change over the historical period (fig. S4). Note that in contrast to the original submission, we have characterised only the anthropogenic component of warming.

In this revision we have extended the scenarios to also account for a decarbonisation of transport and industrial emissions, which are significant sectors alongside energy generation. This is an improvement over the original Davis & Socolow (2014) analysis, following the suggestions of reviewer #3.

Minor comments:

- Before submitting this again the authors need to add line numbers to the document.

Now included.

- All figures are of very low quality (blurry), consider using less compression, or a vector format.

We agree, and supply better quality graphics in this revision.

Reviewer #2 (Remarks to the Author):

Review of 1.5 °C – are we there yet? by C. J. Smith et al

Summary:

A simple carbon-climate model is used in a large ensemble (with varied model parameters) to explore three future carbon emission scenarios in relation to the 1.5 °C maximum warming target set by the Paris Agreement to avoid the most dangerous consequences of anthropogenic warming.

The manuscript finds that we have not yet reached the point at which 1.5 °C becomes impossible. The authors consider both scenarios where fossil fuels are immediately ceased and more realistic scenarios where fossil fuels are phased out as the existing infrastructure is decommissioned and replaced by non-fossil fuel emitting alternatives.

Other recent studies have considered what happens to warming when carbon emissions are suddenly and completely ceased. However, the consideration here of more realistic scenarios makes the findings unique and important, and definitely relevant to the multi-disciplinary audience of Nature Communications.

There are some substantive issues that I would like further clarification on before supporting publication of this manuscript. I feel that clarification on these issues will help readers in the climate modelling field place the findings of this manuscript in context, while also providing additional confidence on the findings for inter-disciplinary readers.

Thank you for the time spent in your thorough and mostly positive review. We agree that application of phase-out scenarios alongside zero-emissions commitments makes our contribution relevant for assessing 21st century mitigation potential.

Therefore, my recommendation is to return to the authors for clarification of the issues raised below.

Substantive points:

Point (1): The interrelationships between the TCR, ECS and ocean ventilation timescales:

The authors define a TCR and ECS for each model simulation within their ensemble. I understand that the authors link the TCR in some way to the ECS (i.e. they are co-varied), based on the statistical link in the CMIP5 ensemble.

The reviewers' understanding is correct: we use a joint-lognormal distribution with correlation coefficient of 0.81, based on CMIP5 models as described in Smith et al. (2017). We have expanded the description in Methods to better inform the reader.

However, the timescales of ocean tracer uptake are also important in setting how the TCR and ECS relate. For example, a system with fast ocean ventilation will have smaller TCR than a system with slower ocean ventilation if the ECS is held constant – since in a system with infinitely fast whole-ocean equilibration for heat and carbon the TCR and the ECS would be the same thing.

To assist the reviewer and others in understanding how the model converts forcing to temperature anomalies we have included a concise overview of the model in the methods section. Equations (2-6) describe how the fast (T_1) and slow (T_2) components of the overall temperature change are proportional to the thermal coefficients q_1 and q_2 . q_1 and q_2 depend on ECS, TCR and the fast and slow (d_1 and d_2) ocean ventilation time constants. See also Smith et al. (2017), eqs. 19 and 20 (q is called c in that reference).

This means that the TCR should be a function of the ECS (which it is in their ensemble) and a function of the two ocean ventilation timescales (which it is not in their ensemble).

Actually, TCR is a function of q_1 and q_2 and d_1 and d_2 , and by Smith et al. (2017) eq. 19 $ECS = F_{2x}(q_1 + q_2)$. So it is q_1 and q_2 , the relative contributions to the fast and slow warming, that are important for driving the model response. However, it is not often known what q_1 and q_2 are, but ECS and TCR are standard diagnostics from climate modelling experiments. q_1 and q_2 can be calculated by inverting the simultaneous equations in Smith et al. (2017) eqs. 19 and 20. We now show this explicitly in equations 4 and 5.

Can the authors justify why they should vary the ocean ventilation timescales independently from TCR/ECS relationship? Surely the TCR should emerge from the ocean ventilation timescales and the ECS?

The reviewer is correct in that there is a weak positive correlation between ECS and d_2 ($r^2=0.26$) or TCR and d_2 ($r^2 = 0.18$), using ECS and TCR values from Forster et al. (2013) and d_2 values from Geoffroy et al. (2013) from the 15 models common to both studies. There is no correlation between ECS/ d_2 and TCR/ d_2 in our constrained ensemble (fig S9). As it would be fairly complex to produce d_2 values that are normally distributed and correlated to ECS variates that are lognormally distributed, since the correlation is weak in CMIP5 models we do not propose to change our input ensemble. However, we make a note of this in the Methods section when describing the distributions.

Or, more likely for their simple model set up, surely the ocean tracer uptake timescales should be co-varied depending on the difference between the TCR and the ECS.

(i.e. In your ensemble design, the difference between the TCR and ECS should be greatest when the ocean ventilation timescales are slowest, and the difference between the TCR and ECS should be smallest when the ocean ventilation timescales are quickest).

In fact, this is what occurs, because q_1 and q_2 do co-vary with the difference in ECS and TCR (eqs. 4 and 5).

Point (2): Using a model with defined TCR to investigate situations in which CO₂ is not increasing at 1% per year.

The Transient Climate Response (TCR) is defined as the warming relative to a preindustrial steady state when atmospheric CO₂ has reached double preindustrial levels, via a pathway of continuously increasing atmospheric CO₂ at an exponential rate (at 1% per year every year).

The simple carbon-climate model (FAIR) used by the authors requires the value of the TCR to be defined. The model then calculates warming over time based, in part, on this defined value of TCR.

However, the authors use their simple carbon climate model to investigate a system that is very different from the continuous exponential increase in atmospheric CO₂.

The whole concept of TCR is undefined (and is therefore not applicable) for a system in which emissions suddenly reduce/cease and atmospheric CO₂ remains steady or falls.

It is unclear (so far as I know) how the emerging value of the TCR in complex climate models relates to the time evolution of surface temperatures in a scenario after emissions are suddenly ceased.

Can the authors provide some clarification as to how the TCR is affecting the evolution of surface temperatures in their simple carbon-climate model simulations after emissions cease? (Presumably this is in part defined by the equations used by the model?).

And can they then discuss whether influence of the TCR on warming in a situation in which the TCR is undefined is reasonable?

At the very least the authors should present an additional figure in the supplementary material, similar to figure S1, showing the variation in committed warming with the model values of TCR and ECS.

Thank you for your detailed comments.

We understand that TCR is an emergent property of climate models run under a highly idealised and unrealistic forcing pathway (i.e. a yearly 1% compound increase in CO₂

concentrations). Therefore we fully appreciate the reviewer's concerns with the apparent use of TCR to drive the model response.

However, it is not the TCR itself that affects the evolution of surface temperature in FAIR. We use TCR and ECS to derive coefficients q_1 and q_2 which govern the fast and slow contributions to temperature change from an input of forcing as described above. Without TCR and ECS to "anchor" these equations to, there would be more than two unknowns for a system of two equations, and there would be an infinity of solutions.

We include fig. S9, which shows the variation in year-2100 temperature with each of ECS, TCR, total ERF in 2018, aerosol ERF in 2018, thermal response time constant of the deep ocean (d_2) and ERF from a doubling of CO₂ (F_{2x}). This is a much expanded version of the old fig. S1.

It would also be good to see a supplementary table that showed the correlation of different model input parameters to the committed warming (e.g. the R^2 correlation coefficient for each model input property versus committed warming).

Figure S9 now includes plots of regression relationships between each parameter, and the r^2 values are displayed in each subplot.

This would help the reader to assess what is leading to the variation in future warming in the model ensembles, and fully understand the results of this study.

We trust that this is now more clear. In addition, the results of the variance-based sensitivity analysis have been incorporated into the main manuscript and figure 4, which shows that the largest contributions to future temperature change in these low-end forcing scenarios are the present-day aerosol forcing and the ECS.

Point (3): Relationship to existing literature

There are existing studies in the literature that look at how warming continues (or not) after emissions are suddenly ceased, e.g.:

Ehlert, D., & Zickfeld, K. (2017). What determines the warming commitment after cessation of CO₂ emissions? *Environmental Research Letters*, 12, 015002.

Frölicher, T. L., Winton, M., & Sarmiento, J. L. (2014). Continued global warming after CO₂ emissions stoppage. *Nature Climate Change*, 4, 40–44.

Williams, R. G., Roussenov, V., Frölicher, T. L., & Goodwin, P. (2017). Drivers of continued surface warming after cessation of carbon emissions. *Geophysical Research Letters*, 44.

The results of this study ought to be compared to this existing literature. Does the simple model ensemble used here behave consistently with the complex model simulations considered in Frölicher et al (2014), or the complex model simulation and simple model ensemble considered in Williams et al (2017)?

The range of committed warming estimates from these studies is quite diverse. We have included the following review in the introduction in order to highlight these ranges of estimates:

In previous studies, the long-term zero-emissions commitment ranges from around -0.4 to +0.9°C, and is sensitive to the carbon cycle response^{1, 2}, ocean heat uptake³, magnitude and pathway of historical warming⁴, and whether or not non-CO₂ forcing is included^{4, 5}. When non-CO₂ forcing is included, setting emissions to zero leads to an initial rapid warming, associated with the removal of short-lived cooling aerosols, followed by a slower decline and stabilisation of temperatures driven by a reduction in the concentrations of short- and long-lived greenhouse gases¹.

References for this paragraph (numbers do not correspond to main paper):

1. Matthews HD, Zickfeld K. Climate response to zeroed emissions of greenhouse gases and aerosols. *Nat Clim Change* **2**, 338-341 (2012).
2. Frölicher TL, Winton M, Sarmiento JL. Continued global warming after CO₂ emissions stoppage. *Nat Clim Change* **4**, 40-44 (2014).
3. Williams RG, Roussenov V, Frölicher TL, Goodwin P. Drivers of Continued Surface Warming After Cessation of Carbon Emissions. *Geophys Res Lett* **44**, 10633-10642 (2017).
4. Ehlert D, Zickfeld K. What determines the warming commitment after cessation of CO₂ emissions? *Environ Res Lett* **12**, 15002 (2017).
5. Samset BH, *et al.* Climate Impacts From a Removal of Anthropogenic Aerosol Emissions. *Geophys Res Lett* **45**, 1020-1029 (2018).

We should point out that our simulations do not correspond to the three studies suggested by the reviewer which all look at idealised experiments using CO₂-only forcing under much higher concentrations than we do in our study. Ehlert and Zickfeld (2017) show that committed warming is much higher under 4xCO₂ than 2xCO₂, but our scenarios peak at around 1.5-1.7xCO₂ (and then decline), so it should not be a surprise that our committed warming is lower.

As discussed in our response to reviewer #1, in contrast to most previous studies (with the exception of Matthews and Zickfeld (2012)) we analyse multi-forcer scenarios. Non-CO₂ forcing can be significant and can have large impacts on carbon budgets (Tokarska and Gillett, 2018; Tokarska et al., 2018). However, to demonstrate the applicability of the model we apply it to the Frölicher et al. (2014) (emulating NCAR CSM1) scenario above. We cannot do this for the Williams et al. (2017) paper, as we do not have the necessary parameters to emulate the GFDL ESM model, and the Ehlert and Zickfeld (2017) experiment requires a yearly adjustment to emissions to achieve a 1% per year concentration increase, so this would be very difficult to emulate.

We repeat a 1000-year experiment with a 1800 GtC pulse against a pre-industrial background, using the emulation of the NCAR CSM1 model in Millar et al. (2017) (note this is not the same AR5-average carbon cycle that we use in this manuscript), and the ECS for this model of 2.0K derived by Joos et al. (2013) (we do not know TCR so estimate it to be 60% of ECS which is a CMIP5 model average). The below plots show the atmospheric CO₂ concentrations and the temperature change derived by FAIR under this model setup. This compares very well to the results in Frölicher et al. (2014) (blue curves in their figs 1a and 1b).

Figure B: Atmospheric CO₂ concentrations emulated from a 1800 GtC pulse with the NCAR CSM1 model emulated by FAIR.

Figure C: Temperature change since pre-industrial for a 1800GtC pulse emission against a pre-industrial background using FAIR emulation of the NCAR CSM1 model.

If the simple model used here behaves consistently with other studies when emissions are suddenly ceased, then this would provide additional confidence for the findings of this study in the unique and important new scenarios where emissions are phased out as infrastructure is decommissioned.

We trust the above plots, along with our comparison to the RCP scenarios (see below) are satisfactory to demonstrate that the model produces plausible responses to emissions.

Point (4): Future projections from this model ensemble for the standard scenarios

What are the future projections for this simple model ensemble for a standard RCP scenario (e.g. RCP8.5 or RCP2.6), and how do these compare with existing model projections?

It would be good to see how this particular ensemble does, compared to existing results, when projecting warming for a standard future scenario. This would again provide additional confidence in the results of this study, and provide more context for how climate scientists should interpret the results presented here with existing results in the literature.

Note that I do not think that disagreement with existing projections for a standard scenario would mean that the results of this study were less important. Just that, for other climate scientists to interpret the results of this study in context, we should see how this model ensemble does for scenarios we are familiar with.

This context is especially important for interpreting the probabilistic language of the manuscript (see minor point below). Therefore, could a supplementary figure be shown where the simple model ensemble projections for a standard RCP scenario are compared to existing projections from other model ensembles in the literature.

Thank you for your comment here. We agree entirely that comparison with the RCP scenarios are necessary to provide confidence that the model is producing realistic projections.

Shown below in figure D are comparisons to the four RCP scenarios. These figures are taken from the FAIR v1.2 model description paper which is in its second review in Geoscientific Model Development and may be accepted shortly (the version in Smith et al. (2017) shows the output for FAIR v1.1, under which some of the relationships for aerosol and ozone forcing are different). It can be seen that the temperature projections for RCP2.6, RCP4.5 and RCP6 agree very well with the emissions-based MAGICC projections (used in AR5) for these scenarios. Our 2100 projections for RCP8.5 are around 0.5K lower than in MAGICC, which we attribute to a lower ensemble average climate sensitivity in our study (based on observational constraints) compared to the MAGICC ensemble. For this study, the low end projections are more applicable, and FAIR reproduces the MAGICC estimates. We make a comment on this in the Methods section.

Figure D: temperature response of the RCP pathways in FAIR v1.2. The bars on the right compare FAIR 2081-2100 mean temperature against emissions-based estimates in MAGICC.

Point (5): Atmospheric CO₂ at equilibrium

The simple model used here applies an impulse response function to determine atmospheric CO₂ over time in response to carbon emissions. The way this is described in this study seems to suggest (to me) that the final atmospheric fraction of CO₂ is fixed by the impulse response function, and is the same for any size of carbon emissions.

However, due to the non-linearity of ocean carbonate chemistry, the final atmospheric fraction of CO₂ varies with emission size.

(As the cumulative emission size grows the final atmospheric fraction of emitted CO₂ also grows, until emissions reach around 5000PgC. This means that the final atmospheric fraction for a cumulative emission size of 500PgC is less than the final atmospheric fraction for a cumulative emission size of 5000PgC, e.g. see Goodwin, P., R. G. Williams, M. J. Follows, and S. Dutkiewicz (2007), Ocean-atmosphere partitioning

of anthropogenic carbon dioxide on centennial timescales, *Global Biogeochem. Cycles*, 21, GB1014, doi:10.1029/2006GB002810).

If my reading of how the model used here works is correct, can the authors reveal what the final atmospheric fraction of CO₂ is from their impulse response function, and whether this is consistent with a cumulative emission size similar to the size implied in their scenarios (something like 600 to 700PgC)?

I may of course have read this wrong, and the impulse response function used by the model may account for the non-linearity of carbonate chemistry. In which case, can the authors just clarify how this is done in the methods/supplementary section.

We are not sure what is meant by final atmospheric fraction, but this is probably not relevant (or if it is, it will be emissions-path dependent) as the model includes biogeochemical feedbacks for reducing efficiency of carbon sinks with both temperature and cumulative CO₂ emissions.

The uptake of carbon by the land and ocean sinks is modelled by adjusting the four time constants of atmospheric decay of CO₂ as parameterised by the integrated airborne fraction over a time horizon of 100 years. As more carbon is added to the atmosphere or the global temperature increases, the 100-year integrated airborne fraction increases, which increases the time constants, and this atmospheric CO₂ is slower to be removed. This is described in (Millar et al., 2017; Smith et al., 2017).

We can show how the (instantaneous) airborne fraction of CO₂ (i.e. how much of a given year's emissions remain in the atmosphere) changes over the historical period and into the future under the SSP2-2018-MID scenario in this paper (figure A in this response). After about 30 years of phase out, CO₂ concentrations peak and start falling even though emissions have not yet reached zero which is why a negative airborne fraction is seen (the plot is cut here because as emissions approach zero airborne fraction - defined as delta CO₂ concentrations divide emissions - starts to blow up). Hence, after a historical airborne fraction of around 0.45 (in agreement with AR5), we get a steadily declining airborne fraction under phase out until just before net zero emissions in which the land and ocean sinks absorb more carbon than is put into the atmosphere in a given year.

Minor point:

Point (1): Writing style:

I find the writing of the manuscript in places lacks clarity and context. For example, consider the following statement from the second paragraph of the Main text (beginning "Figure 1 compares..."):

"Figure 1 compares the committed warming from 2018 (panel a) with the committed warming from 2030 when assuming the emissions currently implied by the Nationally Determined Contributions (NDCs) submitted under the Paris Agreement (panel b,...)"

Figure panels 1a and 1b do not show this. What figure 1 panel (a) and (b) do show is projections of committed warming from a particular model, configured in a particular

way to perform ensemble experiments, under these two scenarios. This context is important, and should certainly be included within this paragraph – preferably before referencing Figure 1. Something like “We use a simple climate model (Ref) to perform ensemble experiments (Methods) and project future committed warming for different scenarios. Figure 1 shows our projections of committed warming for ...”.

The reviewer makes a good point. Following extensive revisions to the paper we introduce the model and experiments in previous paragraphs before the discussions of the results in figure 1. We trust that the context is now clearer. Overall we have now been more careful in the manuscript composition.

Elsewhere, there are references to “X% chance” and “X% probability” of warming outcomes for a particular scenario. Again, this language is used without context, and reads as though the authors have constructed definitive probabilistic warming projections. What the authors mean is that “X% per cent of simulations”, or “the X percentile of simulations”, in their ensemble have a particular warming outcome under that scenario.

This is an important distinction, and one that readers should be made aware of explicitly within the main text of the article, without having to read the methods/supplementary material to assess whether the authors have actually shown definitive probabilistic outcomes.

We agree that this may have given the impression that our results were definitive rather than the outcomes under an ensemble of prescribed model and scenario. In cases where we state probabilities, we have changed the language to say for example “66% of ensemble members” rather than “66% probability”.

Reviewer #3 (Remarks to the Author):

Smith et al. tackle the salient question of whether committed warming is in conflict with the international target of avoiding a 1.5degC in mean temperatures. Although I am sympathetic to their arguments, I have serious misgivings about the manuscript that I think should prevent it from being published without major revisions.

Thank you for your detailed and fair review of this paper. We hope to have addressed your concerns as detailed below.

My overarching criticism is that the authors' main conclusion—that the 1.5deg target “still remains geophysically possible with ambitious emission reduction”—is not well supported by their methods. This could in principle be remedied by either much clearer descriptions of methodological limitations or by using more rigorous methods, but I suspect the former approach would take a lot of punch out of the manuscript (perhaps so much that Nature Communications would not be interested in publishing it), and the latter would entail substantial further work which the authors might not want to do.

Following fair criticisms from the present reviewer and others, we have substantially revised the manuscript to explain the model and inputs in more detail.

Much of the detail of the model relies upon the Smith et al. (2017) paper and also on Millar et al. (2017) so we try to avoid too much overlap with these papers (noting that Smith et al. was not published when this paper was first submitted, so it is completely understandable that the reviewer may need more explanation of how the model works). As described in response to other reviewers, Smith et al. (2017) is under its second review in GMD and should hopefully be accepted shortly. In any case, more detail has been added to the main manuscript in the Methods section.

In this revision we have improved the assumptions behind the scenarios, including phase-out scenarios from the non-energy emissions as well as energy emissions.

A secondary criticism is that the text itself is also not very well written, with confusing and thin explanations of the methods and data, and discussion that is frequently garbled by ambiguous definitions and too-brief methodological descriptions.

The main methodological concerns I have are (1) the climate model and (2) the estimation of committed emissions.

Although the author team includes some very well-respected climate modelers, there is very little validation of this (unpublished) model in the manuscript. The result that most concerns me is that—judging by Fig.1—global mean temperatures begin to decrease within 40 years of a cessation of emissions. Unless these runs include large net negative emissions (not clear), this seems contrary to previously published results of at least some full GCMs and EMICs that show approximately stable mean temperatures for several centuries after net-zero emissions are achieved (see, e.g., Hadley and UVic results in Matthews and Weaver, NatGeo, 2010). I could certainly be persuaded that the model is correct and accurately reflects the latest and greatest CMIP models, but the dearth of validation combined with the importance and contradictory nature of this result raise red flags.

We should have been more clear in our description that the scenarios do not contain negative emissions. New supplementary figures 1 and 2 detail the emissions scenarios used.

In Matthews & Weaver, it appears - although this is not stated explicitly in the text - that their simulations were performed with CO₂ as the only forcing agent, and the sudden increase in temperatures seen from a fast removal of aerosols is not present. As touched upon in the response to the second reviewer, these results are not directly comparable to our study. We provide a similar analysis to Matthews and Zickfeld (2012) in a new supplementary figure 6. In all cases the behaviour of the carbon cycles are important. The reviewer will note that in Matthews & Weaver, the Bern2.5CC model does show a gradually decreasing surface temperature for the zero emissions commitment. Furthermore, as Matthews and Zickfeld (2012) show, temperature stabilisations at lower values than at present are possible with a low present day aerosol forcing. We adopt a low value of aerosol forcing based on estimates from AR5.

Our carbon cycle is set up to emulate the simple model in AR5, which is based on Joos et al. (2013). It is compared to the response of EMICs and ESMs in Millar et al. (2017) (their fig. 4) which shows these complex models can be emulated.

At least as important is what amounts to a concerning mischaracterization of “phase-out” scenarios in the text. Throughout the main text, the authors present these scenarios so as to suggest an update of the bottom-up analysis of global fossil fuel infrastructure in Davis et al. 2010., e.g. discussing power plant lifetimes. However, the supplementary materials suggest that what they have actually done is to apply the various rates of emissions reductions in the previously published analyses to current fossil emissions. This method, however, will drastically underestimate committed emissions because the age distribution of fossil fuel burning infrastructure has changed substantially since 2010—e.g., many new power plants have been built, particularly in China and the U.S. (see Davis and Socolow, 2014) which have decreased the rate at which emissions will decline when a given lifetime is assumed.

Thank you for this suggestion. Based on this we have updated our analysis and used the phase out rates assumed by Davis and Socolow (2014). We are not aware of a more up-to-date reference that details power plant lifetime and energy mix.

The climate model results seems to represent a low-end estimate of future warming and the committed emissions are calculated in such a way that they underestimate future emissions. Neither of these possible biases are conservative to the authors’ very strong claims that 1.5deg is geophysically possible with ambitious mitigation efforts. For this reason, as I stated earlier, I do not support publication of the manuscript as written and recommend substantial revisions either to the claims or the methods (or at least a clear and robust justification of the current methods).

We show in figure D above that the climate model itself does not provide low estimates of warming for RCP2.6, which is the RCP most relevant for these scenarios.

The suggestion that the original Davis et al. (2010) reference was not up to date prompted us to think about phasing out emissions from other sectors aside from energy, so we now include transport, industry and (a very simplified approach for) agriculture. We thank the reviewer for this suggestion as we believe this had led to a more robust set of emissions assumptions.

Minor comments:

-I think it’d be easier to read Fig. 1 if the various commitments were separated into other panels.

We concur it may be easier to read, but feel it would be harder to compare the different commitments, so we have decided to keep with the three commitments on each panel.

-use “natural” and “technical” lifetime for infrastructure. I don’t think either of these is quite right. “Historical” or perhaps “design” or “expected” lifetime would all be preferable. But at least be consistent.

We still use “infrastructural commitment” to describe these scenarios in one word, but now talk about “expected” lifetime. The scenarios themselves have been renamed (see table 1).

-discussion of SSPs and "NDC-conditional trajectories" are especially opaque and could do with a sentence or two more explanation.

Thank you for this suggestion - we agree that we did not spend enough time on this description. We now define the SSP scenarios and conditional/unconditional NDCs in more detail (with references).

References:

- H. D. Matthews, A. J. Weaver, Committed climate warming. *Nature Geoscience* 3, 142-143 (2010).**
- S. J. Davis, K. Caldeira, H. D. Matthews, Future CO₂ Emissions and Climate Change from Existing Energy Infrastructure. *Science* 329, 1330-1333 (2010).**
- S. J. Davis, R. H. Socolow, Commitment accounting of CO₂ emissions. *Environmental Research Letters*, (2014).**

References used in this response:

- Davis, S.J., Caldeira, K. and Matthews, H.D. 2010. Future CO₂ Emissions and Climate Change from Existing Energy Infrastructure. *Science*. **329**(5997), pp.1330-1333.
- Davis, S.J. and Socolow, R.H. 2014. Commitment accounting of CO₂ emissions. *Environmental Research Letters*. **9**, p084018.
- Ehlert, D. and Zickfeld, K. 2017. What determines the warming commitment after cessation of CO₂ emissions? *Environ. Res. Lett.* **12**, p15002.
- Forster, P.M., Andrews, T., Good, P., Gregory, J.M., Jackson, L.S. and Zelinka, M. 2013. Evaluating adjusted forcing and model spread for historical and future scenarios in the CMIP5 generation of climate models. *Journal of Geophysical Research-Atmospheres*. **118**, pp.1139-1150.
- Frölicher, T.L., Winton, M. and Sarmiento, J.L. 2014. Continued global warming after CO₂ emissions stoppage. *Nature Climate Change*. **4**, pp.40-44.
- Geoffroy, O., Saint-Martin, D., Olivié, D.J.L., Voldoire, A., Bellon, G. and Tytéca, S. 2013. Transient Climate Response in a Two-Layer Energy-Balance Model. Part I: Analytical Solution and Parameter Calibration Using CMIP5 AOGCM Experiments. *J. Climate*. **26**, pp.1841-1857.
- Jacobson, M.Z. and Delucchi, M.A. 2011. Providing all global energy with wind, water, and solar power, Part I: Technologies, energy resources, quantities and areas of infrastructure, and materials. *Energy Policy*. **39**, pp.1154-1169.
- Joos, F., Roth, R., Fuglestedt, J.S., Peters, G.P., Enting, I.G., von Bloh, W., Brovkin, V., Burke, E.J., Eby, M., Edwards, N.R., Friedrich, T., Frölicher, T.L., Halloran, P.R., Holden, P.B., Jones, C., Kleinen, T., Mackenzie, F.T., Matsumoto, K., Meinshausen, M., Plattner, G.-K., Reisinger, A., Segschneider, J., Shaffer, G., Steinacher, M., Strassmann, K., Tanaka, K., Timmermann, A. and Weaver, A.J. 2013. Carbon

- dioxide and climate impulse response functions for the computation of greenhouse gas metrics: a multi-model analysis. *Atmos. Chem. Phys.* **13**, pp.2793-2825.
- Matthews, H.D. and Zickfeld, K. 2012. Climate response to zeroed emissions of greenhouse gases and aerosols. *Nat. Clim. Change.* **2**, pp.338-341.
- Millar, R.J., Nicholls, Z.R., Friedlingstein, P. and Allen, M.R. 2017. A modified impulse-response representation of the global near-surface air temperature and atmospheric concentration response to carbon dioxide emissions. *Atmos. Chem. Phys.* **2017**, pp.7213-7228.
- Smith, C.J., Forster, P.M., Allen, M., Leach, N., Millar, R.J., Passerello, G.A. and Regayre, L.A. 2017. FAIR v1.1: A simple emissions-based impulse response and carbon cycle model. *Geoscientific Model Development Discussions.* **2017**, pp.1-45.
- Tokarska, K.B. and Gillett, N.P. 2018. Cumulative carbon emissions budgets consistent with 1.5 °C global warming. *Nature Climate Change.* **8**(4), pp.296-299.
- Tokarska, K.B., Gillett, N.P., Arora, V.K., Lee, W.G. and Zickfeld, K. 2018. The influence of non-CO₂ forcings on cumulative carbon emissions budgets. *Environmental Research Letters.* **13**(3), p034039.
- Williams, R.G., Roussenov, V., Frölicher, T.L. and Goodwin, P. 2017. Drivers of Continued Surface Warming After Cessation of Carbon Emissions. *Geophysical Research Letters.* **44**, pp.10633-10642.

Reviewer #2 (Remarks to the Author):

Review of 1.5C – are we there yet by Smith et al (resubmitted version).

The authors have made concerted efforts to address concerns raised. However, their responses regarding ocean carbon uptake are unconvincing. Indeed, now that I understand better how the FAIR model operates I have serious misgivings as to the applicability of the model, as described, to diagnose carbon emissions. Specifically, I find it entirely acceptable to use a model such as FAIR to investigate warming from prescribed CO₂ pathways, but I have serious concerns about either driving FAIR with or carbon emissions, or diagnosing the carbon emissions from a specified CO₂ pathway.

My concerns arise because the FAIR model does not contain ocean carbonate chemistry, which has a profound and perfectly solvable impact on how emissions relate to atmospheric CO₂. In the construction of the FAIR model, the authors have chosen to ignore the numerous numerical solutions that exist that calculate the impact of ocean carbonate chemistry on the link between emissions and atmospheric CO₂. Instead, the FAIR model lumps ocean and terrestrial carbon uptake together, and uses a single Impulse Response Function to describe the net effect of ocean and terrestrial carbon uptake. The FAIR model does adjust the Impulse Response Function carbon uptake timescales in an attempt to account for carbon-climate feedbacks in the ocean and terrestrial systems. However, the timescales of ocean carbon uptake are not what is changing – the non-linear carbonate chemistry is what is changing. The authors have not demonstrated that this timescale-adjustment has an identical impact as solving ocean carbonate chemistry in all simulations and for all perturbations. Thus, they have not shown that their results would be identical if they did explicitly solve ocean carbonate chemistry. In addition, since ocean and terrestrial carbon uptake operate under fundamentally different and independent processes, the FAIR model's approach of lumping both ocean and terrestrial carbon uptake together may also affect the simulated parameter space they produce.

It is true that in the IPCC 5th Assessment report they use a similar Impulse Response Model for ancillary purposes. The difference here is that the entire conclusions of the paper are drawn from parameter space search using an Impulse Response Function model that does not solve ocean carbonate chemistry. In contrast, none of the findings of the IPCC 5th Assessment report are based solely on models that do not solve ocean carbonate chemistry.

Due to my concerns (listed in more detail below), I am unwilling to support publication of this manuscript until the authors demonstrate that their simulated findings are unaffected by the lack of solving ocean carbonate chemistry, and the lumping together of the ocean and terrestrial carbon uptake when the processes are in fact independent. (i.e. they need to show that identical model results are achieved using a model in which ocean carbonate chemistry is solved, and the processes for terrestrial carbon uptake are solved independently from ocean carbon uptake).

Major concerns:

(1) The treatment of the chemical buffering of carbon dioxide in the ocean.

Unlike many other gasses, such as Oxygen, when carbon dioxide is dissolved in seawater it combines with water to form carbonic acid, and then dissociates into bicarbonate and carbonate ions. The relative amount of Dissolved Inorganic Carbon remaining in each species (CO_2 +carbonic acid, HCO_3 and CO_3) is pH dependent, and the pH is a function of how much carbon has been added. This introduces a chemical buffering into the system, where the greater the amount of carbon added to the system the greater the fraction of CO_2 that will remain in the atmosphere. Roughly speaking, at present conditions around 13 times more carbon will remain as CO_2 in the ocean without dissociating than the pre-existing ratio of un-dissociated to dissociated species. However, that fraction increases as the total amount of carbon added to the air-sea system increases, because of the pH dependence on the chemical buffering.

This chemical buffering effect has been well-known since the first paper on the subject appeared way back in 1956, by Roger Revelle and Hans Suess. The chemical 'buffer factor' describing this phenomenon (approximately equal to 13 at present, but currently rising in the surface ocean) is often known as the 'Revelle' buffer factor because of this.

The FAIR model does not simulate this nonlinear chemical buffering. Instead, the FAIR model attempts to account for the effect of chemical buffering on ocean carbon uptake by altering the ocean carbon uptake timescales.

It is very unusual for model studies to attempt to bypass the ocean buffering of carbon dioxide when simulating ocean carbon uptake (or implicitly simulating ocean carbon uptake as in the case of the FAIR model). I had not appreciated that there was no mechanism to simulate the chemical buffering in the FAIR model – I had simply assumed that some chemical solution was applied in the surface mixed layer, before an impulse response function was applied to subduct the carbon to the deep.

(Note: Applying a chemical solution in the mixed layer and then an impulse response function for sub-surface subduction is the approach applied in Joos et al, 1996 in Tellus. There, they state in the abstract that "the application of response functions based on a pulse increase to atmospheric CO_2 to characterize oceanic uptake, the conventional technique, does not yield a very accurate result due to nonlinearities in the aquatic carbon chemistry". It seems that FAIR has resorted to a pre-1996 method, which was deemed not accurate due to nonlinearities in the carbonate chemistry).

Fundamentally, the chemical buffering is precisely that, a chemical affect. In the FAIR model it is being treated as a physical affect (a change to a physical timescale). However, the physical timescales of ocean carbon uptake can be calculated, and are unaffected by the chemical buffering. First, there is the timescale for the surface mixed layer of the ocean to exchange CO_2 with the

atmosphere, and then there are the timescales for carbon exchange between the surface mixed layer and the sub-surface ocean.

The timescale for carbon dioxide to enter the surface mixed layer and chemically equilibrate with the atmosphere is order 1 year, and is determined by the depth of the surface mixed layer (m) divided by the gas-exchange 'piston velocity' across the air-sea interface (m/s) [which appears as a coefficient in Henry's Law].

While both the depth of the surface mixed layer and the gas exchange piston velocity will vary slightly with climate change, the timescale will remain order 1 year.

Once in the surface mixed layer, carbon dioxide has several additional timescales to enter the sub-surface ocean. These surface to sub-surface ocean carbon exchange timescales are determined by the ventilation timescales with which the surface ocean ventilates the different sub-surface water masses.

Again, while these timescales will be somewhat altered by climate change (because temperature is an active tracer in the ocean), they will not alter on anything like the orders of magnitude with which the FAIR model alters the ocean carbon uptake timescales.

I really don't understand why the authors have constructed the FAIR model in this way. There is no need to introduce a physical hack (a change in a timescale) in an attempt to simulate a well-known chemical phenomenon (carbonate chemistry buffering). Especially when the chemical phenomena itself is so well quantified, and many numerical solutions to the chemical phenomena exist in the published literature [see Zeebe and Wolf-Gladrow, 2001 for a comprehensive reference, but also e.g. Follows et al, 2006 for an efficient numerical scheme, among many others].

A second issue with the FAIR model's approach to approximate chemical buffering by altering the ocean carbon uptake timescales is significant potential for a lack of consistency between the (implicit) ocean heat uptake and the (implicit) ocean carbon uptake. Coincidentally, the ocean surface mixed layer also equilibrates with the atmosphere in terms of heat on an annual timescale. Once in the surface mixed layer, this heat is exchanged with the sub-surface ocean via near-identical ventilation pathways as carbon.

By changing the timescales of carbon uptake, instead of the chemical buffering of carbon into the surface ocean, the FAIR model runs the risk of producing simulations that are internally inconsistent between the uptakes of heat and carbon in the ocean over time. This may significantly affect any findings based on the percentiles of the ensemble-simulations in the FAIR model.

On another note relating to the chemical buffering of carbon dioxide in the ocean, the authors state in their response to reviewers comments that they do not know what is meant by 'final atmospheric fraction' of CO₂, and that it is probably not relevant to their study. However, the final fraction of atmospheric CO₂ is relevant to their study, as it alters how the (implicit) ocean carbon uptake is seen in the FAIR model.

If carbon is emitted into the atmosphere-ocean-terrestrial system, the atmospheric CO₂ level will stabilise on a multi-century timescale as ocean overturning causes the entire ocean to become chemically equilibrated with the atmosphere. The final atmospheric fraction (on a multi-century timescale) varies depending on the cumulative carbon emitted. Due to the chemical buffering of the ocean carbonate system, the final atmospheric fraction increases as cumulative carbon emissions increase. In fact, the final atmospheric CO₂ level is exponentially related to the cumulative carbon emitted for the first 5000 PgC (e.g. see Lenton, 2003 for model simulations of the atmosphere-ocean-terrestrial system; or Goodwin et al, 2007 for a full mathematical explanation for the atmosphere-ocean system).

In the FAIR model, the authors have themselves imposed a final fixed atmospheric CO₂ fraction of carbon emitted of 0.2173, which is applied for any emission size. [Note this means that 21.73 % of carbon remains in the atmosphere after many centuries of ocean and terrestrial uptake in the FAIR model – regardless of emission size]. This fixed atmospheric fraction of CO₂ ignores the effect of chemical buffering of CO₂ in the ocean. The result is a linear increase in atmospheric CO₂ with cumulative emissions on a multi-century timescale in FAIR – when the true effect is an exponential increase in atmospheric CO₂ with cumulative emissions (e.g. Lenton, 2003; Goodwin et al, 2007).

The FAIR model approach of altering the timescale of CO₂ uptake may alter the initial CO₂ uptake in response to rising CO₂ and temperature, but cannot alter the final cumulative CO₂ uptake after many centuries.

Thus, a logical consequence of the FAIR model is that the ocean carbon uptake *must* go wrong at some point for all emissions sizes, other than the precise emission size at which the linear approximation used by FAIR happens to intersect the true exponential curve imposed by ocean chemical buffering. Even then, for the one particular emissions size where it would work out eventually, there is no guarantee that the rate ocean carbon uptake over time will be the same for the FAIR model as it would be if the chemical buffering of the ocean carbonate chemistry system were considered: only that the end total ocean carbon uptake would happen to be the same.

(Also note that the ocean and terrestrial carbon uptake is not explicitly separated in the FAIR model, see point 2 below.)

This means the question is: does the ocean carbon uptake go wrong during the timescale of the experiments presented in this study, in any of the ensemble simulations presented in this study?

To be more precise, for me to support the manuscript, the question becomes: have the authors presented any convincing argument to explain why the altered timescale approach will give the same answer as considering the chemical buffering of the ocean carbonate system for all the simulations presented in this study? And over all timescales presented in this study?

The answer to these questions is no, the authors have not demonstrated this.

Note that the comparison between MAGICC and FAIR shown in the response to reviewers for the RCP scenarios is a false comparison to make when considering the applicability of the FAIR model to explore phenomena in which ocean carbon uptake is important. This is because the warming from a prescribed atmospheric CO₂ pathway is independent of both ocean carbon uptake and the chemical buffering of carbon in the ocean. The problem here is that the FAIR model is being used to make an implicit ocean carbon uptake calculation, and that calculation is critical for the results of this study.

For the current version of the FAIR model, I suggest the authors (and other users of the FAIR model) stick to considering parameters that are unaffected by ocean carbon uptake (e.g. warming from prescribed CO₂ pathways). Alternatively, they may like to take advantage of the numerous existing studies and numerical schemes in the literature on chemical buffering of CO₂ in the ocean, and incorporate these processes into the FAIR model.

This lack of chemical buffering in the FAIR model calls into question the results of previous studies in which the FAIR model has been used to calculate ocean carbon uptake over time (or rather where the results of previous studies are sensitive to the implicitly calculated ocean carbon uptake over time in the FAIR model). This includes occasions when a single run of the FAIR model appears consistent with a model containing explicit ocean carbonate chemistry, but the FAIR model is then used to explore parameter space. There is no reason to suspect that the timescale-hack approach employed in the FAIR model will have the same sensitivities over parameter space as the true ocean chemical buffering (and independently operating terrestrial responses), even if one single simulation of FAIR appears to emulate a model that contains the chemical buffering for a particular perturbation. If any of the parameter space is affected by the difference between the timescale-hack and actual chemical buffering of ocean CO₂, then any percentile results (e.g. 66% of simulations show ... about carbon emissions/ocean carbon uptake) generated by the FAIR model are unreliable.

2 The treatment of terrestrial carbon uptake

The terrestrial carbon uptake is affected by temperature and CO₂, but via completely different processes to the ocean carbon uptake. These processes include CO₂ fertilisation, soil carbon residence time and the temperature sensitivity of global net primary productivity.

The fact that the FAIR model applies the same timescale-shift to the implicit terrestrial carbon uptake as the implicit ocean carbon uptake is worrying. The temperature and CO₂ responses of the ocean and terrestrial carbon sinks are fundamentally different in nature.

The fact that the CO₂ fertilisation and soil carbon residence time respond to temperature and CO₂ independently to ocean carbon uptake in reality, but are lumped together in the FAIR model, also may affect the parameter space results, and the likelihood of remaining under 1.5 °C for any particular pathway. This is because the (implicit) ocean and terrestrial carbon uptake are automatically correlated in the FAIR ensemble, but are independent in reality.

References:

Follows, M. J., S. Dutkiewicz, and T. Ito (2006), On the solution of the carbonate system in ocean biogeochemistry models, *Ocean Modelling*, 12,290– 301.

Goodwin et al (2007) Ocean-atmosphere partitioning of anthropogenic carbon dioxide on centennial timescales, *Global Biogeochemical Cycles* 21, doi:10.1029/2006GB002810.

Joos, F., M. Bruno, R. Fink, U. Siegenthaler, T.F. Stocker, C. Le Quere & J.L. Sarmiento, (1996) An efficient and accurate representation of complex oceanic and biospheric models of anthropogenic carbon uptake, *Tellus* 48B, p397-417.

Lenton, T. (2003) *Climate Change to the End of the Millennium*, *Climatic Change* 76, pp7-29.

Zeebe and Wolf-Gladrow (2001) *CO₂ in Seawater: Equilibrium, Kinetics, Isotopes*, Elsevier, pp360.

Reviewer #3 (Remarks to the Author):

The revised manuscript by Smith et al. is substantially improved over the original submission. I particularly appreciate the plain language explanations of different types of geophysical inertia, which I think will be quite useful to readers who may have wrong ideas about the climate system and the inevitability of future warming.

However, I think it requires further revisions, as described below.

I am still surprised by the magnitude of temperature decline by the end of the century in many of the paper's simulations, and see that other reviewers were similarly surprised. I don't find references to discussion papers terribly convincing, and hope that the other reviewers have commented further regarding the suitability and trustworthiness of the FAIR model as used in this study. Given the importance of present-day aerosol forcing to the findings, I think it is important that the paper expand its discussion to include more than just a reference to AR5. Where do estimates of present-day aerosol forcing come from? Do different methods result in systematically different estimates? Can the authors use their results to suggest a rough W/m^2 threshold of present-day forcing that may determine our ability to avoid 1.5degC of warming in different emissions cases? Not only might that be a useful quantity for aerosol researchers, it would make it clear that the top line results of this paper are dependent on an uncertain parameter.

The phase-out scenarios are better than in the prior version, but the main text does a poor job explaining that these scenarios are not bottom-up (i.e. unit-level) projections, but rather rough estimates of emissions reductions possible based on prior work. I think it is important to clarify this. A good way to do this would be to report the mean annual mitigation rate inferred under Fast, Mid, and Slow phase-out scenarios. I believe this is essentially what the authors are calculating from previous studies and applying in a stylized way to estimate the future rate of decline of present-day emissions. For context, the actual rate could also be reported for the past decade.

Citations to papers by Mark Jacobson are carrying a lot of weight in lines 52-53, where they support the proposition that it "may be technically and economically possible [to transition to a zero carbon energy system]." This is highly controversial, and a diverse group of energy modelers have argued that Jacobson's work demonstrates no such thing (Clack et al., 2017). Regardless of which side of the debate the authors are on, it seems important to more explicitly acknowledge the controversy.

Regarding the phase-out of agricultural emissions over the remainder of the century, this is in line with the baseline in the SSPs. However, it is not clear to me if such a phase-out implies large-scale deployment of negative emissions technologies, which many analysts might deem fantastical. To the extent their conclusions may depend on such fantasy, the authors should be clear and discuss. There is a growing literature on this subject, including several important pieces in a recent volume of ERL.

Finally, I encourage the authors to re-read their text carefully, as there remain worrisome errors of language. For example, this first sentence of the “Summary and outlook” section indicates that “limiting warming to 1.5degC...occurs in only 10% of ensemble members...” This is the opposite of what the authors conclude.

We first thank both reviewers again for re-evaluating our manuscript. We took the opportunity to re-run the simulations with the finalised, published configuration of the FAIR model, version 1.3, in Smith *et al.* (2018). Results have changed slightly, but not materially, owing to a small adjustment in the black carbon forcing between versions 1.2 and 1.3 of the model.

For the variance-based sensitivity analysis in Figure 4, we increased the number of samples to 2500 per parameter and did not analyse second-order interactions to make 50000 samples in total. Again, results have changed slightly, but not materially.

Finally, to avoid ambiguity in the Integrated Assessment Modelling community where a model named FAIR exists, we refer to the model in this manuscript as FaIR.

Reviewer #2 (Remarks to the Author):

Review of 1.5C – are we there yet by Smith et al (resubmitted version).

The authors have made concerted efforts to address concerns raised. However, their responses regarding ocean carbon uptake are unconvincing. Indeed, now that I understand better how the FAIR model operates I have serious misgivings as to the applicability of the model, as described, to diagnose carbon emissions. Specifically, I find it entirely acceptable to use a model such as FAIR to investigate warming from prescribed CO₂ pathways, but I have serious concerns about either driving FAIR with or carbon emissions, or diagnosing the carbon emissions from a specified CO₂ pathway.

Thank you for the time spent in reviewing our manuscript a second time and providing a detailed commentary. We also thank the reviewer for taking time to understand how the FaIR model works. We respond point-by-point to your comments below and hope to satisfy your concerns that FaIR can indeed model atmospheric CO₂ pathways from input emissions.

My concerns arise because the FAIR model does not contain ocean carbonate chemistry, which has a profound and perfectly solvable impact on how emissions relate to atmospheric CO₂. In the construction of the FAIR model, the authors have chosen to ignore the numerous numerical solutions that exist that calculate the impact of ocean carbonate chemistry on the link between emissions and atmospheric CO₂. Instead, the FAIR model lumps ocean and terrestrial carbon uptake together, and uses a single Impulse Response Function to describe the net effect of ocean and terrestrial carbon uptake. The FAIR model does adjust the Impulse Response Function carbon uptake timescales in an attempt to account for carbon-climate feedbacks in the ocean and terrestrial systems. However, the timescales of ocean carbon uptake are not what is changing – the non-linear carbonate chemistry is what is changing. The authors have not demonstrated that this timescale-adjustment has an identical impact as solving ocean carbonate chemistry in all simulations and for all perturbations. Thus, they have not shown that their results would be identical if they did explicitly solve ocean carbonate chemistry. In addition, since ocean and terrestrial carbon uptake operate under fundamentally different and independent processes, the

FAIR model's approach of lumping both ocean and terrestrial carbon uptake together may also affect the simulated parameter space they produce.

It is true that FaIR does not explicitly account for ocean carbonate chemistry or an explicit separation of land and ocean carbon response. However, we show that this does not prevent FaIR from being able to simulate carbon dioxide concentration and temperature response to emissions pathways over the historical period and 21st century, the latter of which is the focus of the paper. The best evidence we can present for this is comparisons to models which do include ocean carbonate chemistry. We refer to several previous publications and provide some new examples in our response here.

As described in Millar *et al.* (2017) and Jenkins *et al.* (2018), the carbon cycle routine in FaIR can be tuned to emulate the range Earth System models in CMIP5 by varying three parameters (a subset of the total): the pre-industrial airborne fraction, and two feedback terms being the sensitivity of airborne fraction to total carbon resident in the land and ocean sinks, and sensitivity of airborne fraction to temperature. These are respectively r_0 , r_C and r_T in the notation of Millar *et al.* (2017). Jenkins *et al.* (2018) assumes that the ratio r_C/r_T is constant, reducing the number of free parameters to two, but we retain independence between them in this paper.

After fitting r_0 , r_C and r_T values to CMIP5 models, we create distributions of these parameters informed by these samples. As shown by Jenkins *et al.* (2018) fig. 2a, FaIR replicates the emissions-concentrations relationships from CMIP5 earth system models well for RCP8.5, RCP2.6 (when non-CO₂ forcing is prescribed) and 1%CO₂ using just fitted values of r_0 and r_C/r_T .

When non-CO₂ emissions are also specified, the RCP scenarios that are run from MAGICC are well simulated (Smith *et al.* 2018). MAGICC does include an explicit dissolved inorganic carbon formulation and terrestrial/ocean carbon separation (described in Meinshausen *et al.* (2011), appendix A). Agreement between FaIR and MAGICC is good for RCP2.6 which is most important for the analysis in this paper. Although emulating a more complex, but still simple, model like MAGICC does not validate FaIR, it gives confidence that it is behaving sensibly for the range of future scenarios it is targeted for.

There are a number of reasons that we use FaIR for the analysis here instead of MAGICC. Firstly we know from initial comparisons that FaIR runs an order of magnitude faster than MAGICC. Secondly, the number of free parameters is much smaller in FaIR – there are only 20, 18 of which are varied in this study, compared to 82 in MAGICC. This facilitates a much more tractable sensitivity analysis. Thirdly, FaIR is based on more up-to-date knowledge (AR5 or newer). We are aware that MAGICC7 is in development, but the most recent available version of MAGICC (MAGICC6) has its science basis in AR4. Fourth, one advantage of FaIR over MAGICC is the sampling of carbon cycle feedbacks are not limited to the set of models that it has been calibrated for. For example, FaIR can sample the space of r_0 , r_C and r_T to fill in the gaps between models where emulations of these parameters exist, whereas MAGICC is limited to using one of 10 carbon cycle models that their model has emulated. Finally, although agreement is good for RCP2.6, we have noticed some important differences between the models for scenarios used in the IPCC Special Report on

1.5°C, related to the rate of present day warming, which is an important consideration for low-end forcing scenarios (see Leach *et al.* (2018)).

The conclusion is we claim that FaIR simulates the long term atmospheric airborne fraction evolution well. This is demonstrated in Millar *et al.* (2017) with a comparison of the impulse response function to a suite of carbon cycle models, and with further examples below.

It is true that in the IPCC 5th Assessment report they use a similar Impulse Response Model for ancillary purposes. The difference here is that the entire conclusions of the paper are drawn from parameter space search using an Impulse Response Function model that does not solve ocean carbonate chemistry. In contrast, none of the findings of the IPCC 5th Assessment report are based solely on models that do not solve ocean carbonate chemistry.

Global temperature potential is one metric from AR5 that uses an impulse response model in its calculation. There is also a precedent for publishing papers on committed warming with even simpler models than ours, e.g. Mauritsen and Pincus (2017). We also reiterate that FaIR does well-replicate models that solve ocean carbonate chemistry with the appropriate parameter selection, namely ESMs and EMICs (and MAGICC).

Due to my concerns (listed in more detail below), I am unwilling to support publication of this manuscript until the authors demonstrate that their simulated findings are unaffected by the lack of solving ocean carbonate chemistry, and the lumping together of the ocean and terrestrial carbon uptake when the processes are in fact independent. (i.e. they need to show that identical model results are achieved using a model in which ocean carbonate chemistry is solved, and the processes for terrestrial carbon uptake are solved independently from ocean carbon uptake).

We believe that we can demonstrate evidence in this response (and by reference to previous papers) that FaIR can simulate the behaviour of more complex earth system models over the range of scenarios that are relevant to the question being asked here.

Major concerns:

(1) The treatment of the chemical buffering of carbon dioxide in the ocean.

Unlike many other gasses, such as Oxygen, when carbon dioxide is dissolved in seawater it combines with water to form carbonic acid, and then dissociates into bicarbonate and carbonate ions. The relative amount of Dissolved Inorganic Carbon remaining in each species (CO₂+carbonic acid, HCO₃ and CO₃) is pH dependent, and the pH is a function of how much carbon has been added. This introduces a chemical buffering into the system, where the greater the amount of carbon added to the system the greater the fraction of CO₂ that will remain in the atmosphere. Roughly speaking, at present conditions around 13 times more carbon will remain as CO₂ in the ocean without dissociating than the pre-existing ratio of un-dissociated to dissociated species. However, that fraction increases as the total amount of carbon added to the air-sea system increases, because of the pH dependence on the chemical buffering.

As the reviewer states: “The greater the amount of carbon added to the system the greater the fraction of CO₂ that will remain in the atmosphere”. This response is explicitly captured by FaIR. We do this in a different way (by adjusting the response time scale) to the method that the reviewer is perhaps most familiar with. Equation 8 of Millar *et al.* (2017) gives

$$iIRF_{100} = r_0 + r_C C_{acc} + r_T T$$

where $iIRF_{100}$ is the 100-year time-integrated airborne fraction. This behaviour approximates the behaviour of EMICs and ESMs. With increasing C_{acc} , which is the total carbon addition to the land/ocean system, the 100-year time integrated airborne fraction increases.

This chemical buffering effect has been well-known since the first paper on the subject appeared way back in 1956, by Roger Revelle and Hans Suess. The chemical ‘buffer factor’ describing this phenomenon (approximately equal to 13 at present, but currently rising in the surface ocean) is often known as the ‘Revelle’ buffer factor because of this.

This is acknowledged by Millar *et al.* (2017): “In more comprehensive models, ocean uptake efficiency declines with accumulated CO₂ in ocean sinks (Revelle and Suess 1957) and uptake of carbon into both terrestrial and marine sinks are reduced by warming (Friedlingstein *et al.* 2006).” The importance of this effect is acknowledged and contributed to formulating the way FaIR models the evolution of land plus ocean carbon uptake efficiency.

The FAIR model does not simulate this nonlinear chemical buffering. Instead, the FAIR model attempts to account for the effect of chemical buffering on ocean carbon uptake by altering the ocean carbon uptake timescales.

The reviewer’s understanding is correct. The increase in accumulated carbon in the (land plus ocean) stocks drives the rate of decay from the four carbon boxes used in FaIR. With increasing cumulative carbon, the concentration anomaly decay time constants get longer, the rate of uptake by the land and ocean declines, and airborne fraction increases.

It is very unusual for model studies to attempt to bypass the ocean buffering of carbon dioxide when simulating ocean carbon uptake (or implicitly simulating ocean carbon uptake as in the case of the FAIR model). I had not appreciated that there was no mechanism to simulate the chemical buffering in the FAIR model – I had simply assumed that some chemical solution was applied in the surface mixed layer, before an impulse response function was applied to subduct the carbon to the deep.

We believe that the most relevant effect of ocean carbon buffering for the research questions being asked here (the increase in airborne fraction with ocean uptake of carbon) is captured by FaIR. The importance of this effect explicitly shaped the simple parametrisation chosen the model, whilst attempting to stay true to a minimal complexity emulator philosophy throughout the model. Within the incorporation of this feedback, FaIR would fail to emulate the response of earth system models as well as it does.

Other recent studies on committed warming do not include any physically-based representation of the effect of ocean buffering of carbon dioxide, for example Mauritsen and Pincus (2017) and Schmidt et al (2017).

(Note: Applying a chemical solution in the mixed layer and then an impulse response function for sub-surface subduction is the approach applied in Joos et al, 1996 in Tellus. There, they state in the abstract that “the application of response functions based on a pulse increase to atmospheric CO₂ to characterize oceanic uptake, the conventional technique, does not yield a very accurate result due to nonlinearities in the aquatic carbon chemistry”. It seems that FAIR has resorted to a pre-1996 method, which was deemed not accurate due to nonlinearities in the carbonate chemistry).

The reviewer will be aware that the impulse response function in FaIR is state dependent. If there were no carbon cycle feedbacks in FaIR, then the reviewer would be correct in asserting that the treatment is inaccurate.

Here we show why adding explicit simulation of ocean carbonate chemistry to the simple model will not necessarily improve its response. We replicate experiments from Katavouta *et al.* (2018), which compares a simple model which does include ocean carbonate chemistry to the GFDL-ESM2M Earth System model. The GFDL-ESM2M experiment uses a 1% per year compound CO₂ increase until the time of reaching 2°C at year 99, at which emissions are set to zero and run until year 1000 (originally from Frölicher, Winton and Sarmiento (2014)). We use the tuning specific to GFDL-ESM2M here from Jenkins *et al.* (2018) to show that FaIR is able to replicate the atmospheric CO₂ concentration profile of GFDL-ESM2M (figure (i)a below) following zero CO₂ emissions. FaIR in fact shows a slower drop-off of atmospheric CO₂ concentration following zero emissions than GFDL-ESM2M, so that any long-term commitment estimate would be conservative (at least compared to projections from this particular earth system model).

In Katavouta *et al.* (2018), the authors use their model, which includes ocean carbonate chemistry, to diagnose atmospheric CO₂ concentrations (their fig. 1a, red curves). Although the increasing emissions pathways are different between the simple model and the GFDL-ESM2M shown in fig. 1b (constant 20 GtC in the simple model, 1% compound increase in the ESM), the cumulative emissions at time of net zero are similar (2000 GtC and 1950 GtC) and achieved at a similar time (100 years in simple model, 99 years in GFDL-ESM2M). The model in Katavouta *et al.* (2018) shows an atmospheric response to increasing CO₂ that overestimates the rates of atmospheric CO₂ concentration decline compared to the GFDL-ESM2M with a complete stabilisation after ~500 years (not seen in the ESM in which concentrations continue to decline beyond 1000 years).

FaIR, without its ocean carbon buffering, does model this correctly as seen in fig (i)a.

Figure (i)b below shows how FaIR simulates the temperature from the diagnosed CO₂ concentrations in panel (i)a. Over the 1000-year simulation it can be seen that FaIR produces good long-run estimates of the temperature response from GFDL-ESM2M from both its carbon cycle emulation of GFDL-ESM2M (blue curve) and from GFDL's CO₂ concentrations (pink curve).

Figure (i): (a) comparison of the total atmospheric airborne fraction following a 1% per year compound CO₂ increase until year 99 and then zero emissions until year 1000. Black: cumulative CO₂ emissions; red: airborne carbon in GFDL; blue: airborne carbon in FAIR. (b) Temperature response in GFDL (black), FAIR from FAIR's airborne fraction (blue) and FAIR from GFDL's airborne fraction (pink).

Fundamentally, the chemical buffering is precisely that, a chemical affect. In the FAIR model it is being treated as a physical affect (a change to a physical timescale). However, the physical timescales of ocean carbon uptake can be calculated, and are unaffected by the chemical buffering. First, there is the timescale for the surface mixed layer of the ocean to exchange CO₂ with the atmosphere, and then there are the timescales for carbon exchange between the surface mixed layer and the sub-surface ocean.

FaIR does not attempt to model oceanic CO₂ uptake explicitly. The carbon decay timescales are for atmospheric carbon anomalies as opposed to explicit physical timescales. It is important to not directly associate the decay timescales within FaIR with individual physical processes as they will be the sum of contributions from several distinct physical processes in each case. Nonlinearities in the carbonate chemistry are captured by non-linearities in the efficiency of carbon uptake (or rather, a decline in the efficiency of the decay from each atmospheric box) with increasing carbon. As is highlighted in Joos *et al.* (2013), these time times do show a dependence on the background state of the climate system.

The timescale for carbon dioxide to enter the surface mixed layer and chemically equilibrate with the atmosphere is order 1 year, and is determined by the depth of the

surface mixed layer (m) divided by the gas-exchange ‘piston velocity’ across the air-sea interface (m/s) [which appears as a coefficient in Henry’s Law].

While both the depth of the surface mixed layer and the gas exchange piston velocity will vary slightly with climate change, the timescale will remain order 1 year.

Once in the surface mixed layer, carbon dioxide has several additional timescales to enter the sub-surface ocean. These surface to sub-surface ocean carbon exchange timescales are determined by the ventilation timescales with which the surface ocean ventilates the different sub-surface water masses.

We don’t doubt that all models could be improved to make them more physically realistic. Assuming that the explicitly solving oceanic carbon content would improve the model’s performance (which is not guaranteed), it is certainly guaranteed that extra complexity would be added. We believe that this is not necessarily justified, as demonstrated in our figure (i) above.

Again, while these timescales will be somewhat altered by climate change (because temperature is an active tracer in the ocean), they will not alter on anything like the orders of magnitude with which the FAIR model alters the ocean carbon uptake timescales.

We point out that there is not a physical “ocean uptake timescale” in FaIR as it is not intended to exactly represent different levels of the ocean. It is a modified impulse response model that represents the atmospheric decay of CO₂ as a function of accumulated carbon and temperature.

I really don’t understand why the authors have constructed the FAIR model in this way. There is no need to introduce a physical hack (a change in a timescale) in an attempt to simulate a well-known chemical phenomenon (carbonate chemistry buffering). Especially when the chemical phenomena itself is so well quantified, and many numerical solutions to the chemical phenomena exist in the published literature [see Zeebe and Wolf-Gladrow, 2001 for a comprehensive reference, but also e.g. Follows et al, 2006 for an efficient numerical scheme, among many others].

The model was constructed this way to be a simulator that projects global mean temperatures from source emissions. It was designed to be as simple as possible but no simpler. To model carbon uptake explicitly would require a more complex model, with several interacting boxes in the ocean, biosphere and soil, an atmospheric box and a geological box. Currently, simplicity and fast runtimes are key advantages of FaIR. Having several interacting boxes will lose some of these advantages.

A second issue with the FAIR model’s approach to approximate chemical buffering by altering the ocean carbon uptake timescales is significant potential for a lack of consistency between the (implicit) ocean heat uptake and the (implicit) ocean carbon uptake. Coincidentally, the ocean surface mixed layer also equilibrates with the atmosphere in terms of heat on an annual timescale. Once in the surface mixed layer,

this heat is exchanged with the sub-surface ocean via near-identical ventilation pathways as carbon.

Ocean carbon uptake is not modelled explicitly as we discuss above. Therefore, we do not believe that the timescales within the model will be as closely connected as the physical timescales might be. Fits to the behaviour of CMIP5 models under scenario and idealised experiments show that a range of different combinations of carbon anomaly and thermal anomaly decay timescales are apparent across the ESM ensemble (Geoffroy *et al.* 2013; Joos *et al.* 2013).

By changing the timescales of carbon uptake, instead of the chemical buffering of carbon into the surface ocean, the FAIR model runs the risk of producing simulations that are internally inconsistent between the uptakes of heat and carbon in the ocean over time. This may significantly affect any findings based on the percentiles of the ensemble-simulations in the FAIR model.

The percentiles are drawn from the random sampling across the space of plausible earth system models and radiative forcing uncertainty. FAIR agrees well with the earth system models that it has been trained on, and we presume that no ocean heat uptake/carbon uptake inconsistencies exist in these earth system models. Constraint to observed historical temperatures also introduces an implicit constraint that helps to restrict parameter combinations that are likely to be inconsistent with recent evolution of the climate system.

On another note relating to the chemical buffering of carbon dioxide in the ocean, the authors state in their response to reviewers comments that they do not know what is meant by 'final atmospheric fraction' of CO₂, and that it is probably not relevant to their study. However, the final fraction of atmospheric CO₂ is relevant to their study, as it alters how the (implicit) ocean carbon uptake is seen in the FAIR model.

Thank you for clarifying the meaning of "final atmospheric fraction". As you state below this means 21.73% of the emitted CO₂ stays in the atmosphere quasi-permanently (actually a time constant of 10⁶ years is used).

If carbon is emitted into the atmosphere-ocean-terrestrial system, the atmospheric CO₂ level will stabilise on a multi-century timescale as ocean overturning causes the entire ocean to become chemically equilibrated with the atmosphere. The final atmospheric fraction (on a multi-century timescale) varies depending on the cumulative carbon emitted. Due to the chemical buffering of the ocean carbonate system, the final atmospheric fraction increases as cumulative carbon emissions increase. In fact, the final atmospheric CO₂ level is exponentially related to the cumulative carbon emitted for the first 5000 PgC (e.g. see Lenton, 2003 for model simulations of the atmosphere-ocean-terrestrial system; or Goodwin *et al.*, 2007 for a full mathematical explanation for the atmosphere-ocean system).

The exponential behaviour is in fact seen for the first 5000 PgC emitted in FAIR, in figure (ii) below. Here we have replicated the experiment in Figure 6 of Goodwin *et al.* (2007) using a 10 GtC yearly input of carbon from a pre-industrial background until the cumulative carbon emissions are 500, 1000, 1500, ... 5000 GtC, then emissions ceased and the model

integrated forward for 3000 years with the CO₂ concentration taken at the end of the 3000 years. We don't have a specific fit of the FaIR parameters from the MIT-GCM as we do not have 1%CO₂, 4xCO₂ or any earth system experiments. However, we use the ECS and TCR from the MIT-IGSM2.3 model from Plattner *et al.* (2008), which we assume to be sufficiently similar, and the r_0 , r_T and r_C are fit to produce approximate results to the MIT-GCM results in the 3000-year zero carbon results to Figure 6 in Goodwin *et al.* (2007).

Figure (ii): FaIR approximate emulation of the MIT-GCM from Goodwin *et al.*, 2007. Cumulative emissions (10 GtC per year up until the year of zero emissions) plot against CO₂ concentration after 3000 years of zero emissions.

In the FAIR model, the authors have themselves imposed a final fixed atmospheric CO₂ fraction of carbon emitted of 0.2173, which is applied for any emission size. [Note this means that 21.73 % of carbon remains in the atmosphere after many centuries of ocean and terrestrial uptake in the FAIR model – regardless of emission size]. This fixed atmospheric fraction of CO₂ ignores the effect of chemical buffering of CO₂ in the ocean. The result is a linear increase in atmospheric CO₂ with cumulative emissions on a multi-century timescale in FAIR – when the true effect is an exponential increase in atmospheric CO₂ with cumulative emissions (e.g. Lenton, 2003; Goodwin *et al.*, 2007).

The reviewer is correct with the suggestion that the default partition fraction into the quasi-permanent reservoir (time constant $\tau_0 = 10^6$ years) is 0.2173. It is based on the present-day impulse-response function in AR5. As is shown in figure (iii) below, the atmospheric CO₂ concentration does not stabilise in the zero emissions case for higher emissions scenarios, because the time constant scaling factor α lengthens the decay time of the other carbon boxes. The second longest carbon box has a present-day time constant of $\tau_1 = 394$ years.

For the blue and orange curves in figure (iii), representing 500 GtC and 1000 GtC cumulative emissions, $\alpha \approx 1$ (because temperature and cumulative emissions are not much different to present day), the lifetime of this box remains of the order 400 years, and 3000 years of zero emissions provides enough time for all of the carbon in all except the quasi-permanent box to decay away. In this case, the “final” airborne fraction is indeed about 21.73%. But for the higher emissions scenarios, the atmosphere/ocean has not equilibrated. The α factor after 3000 years of zero emissions in the 5000 GtC emissions case is about 20, so the second longest atmospheric carbon box has a time constant of about 8000 years, and about 70% ($\exp(-\frac{3000}{8000})$) of the carbon in this box is still airborne after 3000 years.

Figure (iii): Transient evolution of atmospheric carbon dioxide and temperature for each 3000-year zero emissions scenario.

FaIR is a flexible model and the user can change the parameter specification to fit the process that they are trying to simulate. There is nothing to stop the user changing the partition coefficients or time constants as a function of emissions if they feel this would result in a better approximation of what they are trying to achieve, or to use a different number of boxes in the impulse-response model to four.

The FAIR model approach of altering the timescale of CO2 uptake may alter the initial CO2 uptake in response to rising CO2 and temperature, but cannot alter the final cumulative CO2 uptake after many centuries.

We show in figures (i) and (iii) that it can if the time horizon of interest (e.g. 3000 years here) is fixed, so we argue that it can. An airborne fraction of 0.2173 following a zeroing of emissions occurs at a time that is several times longer than $\alpha\tau_1$ but several times shorter than $\alpha\tau_0$. As we discuss previously, α is state dependent (and continues to evolve in the future as temperatures change and carbon is outgassed from each box).

In the limit $t \rightarrow \infty$ ($t \gg \alpha\tau_0$) all atmospheric carbon will eventually decay away back to its “pre-industrial” value in FaIR. Although this is not a correct treatment on geological time scales where pre-industrial represents one particular natural epoch, it is far beyond the period of interest for this study and what FaIR was designed to represent.

Thus, a logical consequence of the FAIR model is that the ocean carbon uptake **must go wrong at some point for all emissions sizes, other than the precise emission size at which the linear approximation used by FAIR happens to intersect the true exponential curve imposed by ocean chemical buffering. Even then, for the one particular emissions size where it would work out eventually, there is no guarantee that the rate ocean carbon uptake over time will be the same for the FAIR model as it would be if the chemical buffering of the ocean carbonate chemistry system were considered: only that the end total ocean carbon uptake would happen to be the same.**

We show in figure (ii) that the expected response in FaIR is observed. Figure (i) demonstrates the correct behaviour for carbon uptake over time for a zero emissions commitment.

(Also note that the ocean and terrestrial carbon uptake is not explicitly separated in the FAIR model, see point 2 below.)

We respond to this point separately below.

This means the question is: does the ocean carbon uptake go wrong during the timescale of the experiments presented in this study, in any of the ensemble simulations presented in this study?

Again referring to fig. (i), there is very little discrepancy between FaIR and GFDL-ESM2M for the first 100 years following the zeroing of emissions. This is what we would interpret to be the “timescale of the experiments presented in this study”. We also note that for smaller emissions pulses, more representative of the magnitude of perturbations associated with a zero-emissions experiment, the effect of ocean carbonate feedback effects are expected to be smaller than under high cumulative emissions future experiments.

To be more precise, for me to support the manuscript, the question becomes: have the authors presented any convincing argument to explain why the altered timescale approach will give the same answer as considering the chemical buffering of the ocean carbonate system for all the simulations presented in this study? And over all timescales presented in this study?

We can show this for individual examples over the timescales of interest for experiments performed with earth system models in the literature, e.g. figure (i). We cannot prove that our results will exactly agree with what an earth system model would produce for the scenarios in this paper without directly comparing our results to an earth system model running the same experiments. Time and HPC resource does not permit us to perform these experiments. CMIP6, amongst others, will produce ESM results from low forcing pathways.

The answer to these questions is no, the authors have not demonstrated this.

It is shown for CMIP5 ESMs by Jenkins *et al.* (2018). Millar *et al.* (2017) shows how FaIR emulates carbon-cycle models, Smith *et al.* (2018) shows the comparison to MAGICC (more

on this below), and we provide two additional examples fit to MIT-GCM and GFDL-ESM2M in this response.

Note that the comparison between MAGICC and FAIR shown in the response to reviewers for the RCP scenarios is a false comparison to make when considering the applicability of the FAIR model to explore phenomena in which ocean carbon uptake is important. This is because the warming from a prescribed atmospheric CO₂ pathway is independent of both ocean carbon uptake and the chemical buffering of carbon in the ocean. The problem here is that the FAIR model is being used to make an implicit ocean carbon uptake calculation, and that calculation is critical for the results of this study.

MAGICC does include carbonate chemistry, and from 2005 onwards is run in earth-system (e.g. from emissions of CO₂) mode, which is how the RCP scenarios are derived in the first place. So although decent agreement with MAGICC for the emissions-driven (2005-2100) time period for RCP2.6 does not prove the validity of FaIR for the low-end scenarios here, it does at least suggest that FaIR is producing similar results to a well-known and established model used by the community, as shown by Smith *et al.* (2018).

If the reviewer is more concerned about the behaviour in zero-emissions cases rather than declining emissions cases, we refer to our example in figure (i).

For the current version of the FAIR model, I suggest the authors (and other users of the FAIR model) stick to considering parameters that are unaffected by ocean carbon uptake (e.g. warming from prescribed CO₂ pathways). Alternatively, they may like to take advantage of the numerous existing studies and numerical schemes in the literature on chemical buffering of CO₂ in the ocean, and incorporate these processes into the FAIR model.

We have provided several examples where FaIR is useful for the time period and scenarios of interest. It would be useful if the reviewer could demonstrate how the model does not exhibit the correct behaviour for experiments described in this paper.

The FaIR model is available here: <https://github.com/OMS-NetZero/FAIR/>. The version used in this paper is v1.3.4. An online interactive version can be accessed at <https://mybinder.org/v2/gh/OMS-NetZero/FAIR/master?filepath=Example-Usage.ipynb>.

This lack of chemical buffering in the FAIR model calls into question the results of previous studies in which the FAIR model has been used to calculate ocean carbon uptake over time (or rather where the results of previous studies are sensitive to the implicitly calculated ocean carbon uptake over time in the FAIR model).

As the reviewer is aware FaIR is not used to calculate ocean carbon uptake. It does not claim to have this ability and no results have ever been presented in which ocean carbon uptake is quantified. The airborne fraction is the variable of importance, and all previous studies have either quantified this directly (by verifying the airborne fraction is sensible and in line with other models and estimates from the Global Carbon Budget, e.g. Millar *et al.*

(2017)) or indirectly, for example verifying that present-day CO₂ concentrations predicted by the model are in line with observations (e.g. this study, Smith *et al.* (2018)).

This includes occasions when a single run of the FAIR model appears consistent with a model containing explicit ocean carbonate chemistry, but the FAIR model is then used to explore parameter space.

This is one of the advantages of the simplified framework. If a wide range of ESMs can be simulated using the pre-industrial airborne fraction and varying the strength of carbon cycle and temperature feedbacks, then we allow “all possible” ESMs to be simulated by selecting these parameters from a distribution. Implausible parameter combinations are filtered out. This is the approach taken for the distributions of ECS and TCR, and follows similar publications, e.g. Meinshausen *et al.* (2009) for the MAGICC model.

There is no reason to suspect that the timescale-hack approach employed in the FAIR model will have the same sensitivities over parameter space as the true ocean chemical buffering (and independently operating terrestrial responses), even if one single simulation of FAIR appears to emulate a model that contains the chemical buffering for a particular perturbation. If any of the parameter space is affected by the difference between the timescale-hack and actual chemical buffering of ocean CO₂, then any percentile results (e.g. 66% of simulations show ... about carbon emissions/ocean carbon uptake) generated by the FAIR model are unreliable.

An alternative here would be to only select the set of 15 earth system models and EMICs that have been emulated by FAIR. This is what is done when MAGICC performs similar studies; MAGICC has a calibration for 10 different model carbon cycles, and one is selected at random for each ensemble member.

This would hamper investigation of the carbon cycle uncertainty, although, we show in figure 4 that the contribution to 21st century committed warming is very small anyway. If we took this approach the reviewer could surely not argue that there would be internal inconsistencies between temperature and heat uptake because these models have been emulated for RCP2.6.

2 The treatment of terrestrial carbon uptake

The terrestrial carbon uptake is affected by temperature and CO₂, but via completely different processes to the ocean carbon uptake. These processes include CO₂ fertilisation, soil carbon residence time and the temperature sensitivity of global net primary productivity.

We agree, and this actually be a useful future development. However, what we know from the Joos *et al.* (2013) experiments already is that it isn't as simple as just assuming that only one or two of the impulse-response timescales are associated exclusively with ocean saturation effects and thus only those need to be scaled. Adding significant complexity to the FAIR carbon cycle would be a substantial philosophical change away from a simple minimal complexity impulse response based emulator that, with the dimensionality it already has, can

be demonstrated to be sufficient to accurately emulate ESMs under high and low future emissions scenarios.

The fact that the FAIR model applies the same timescale-shift to the implicit terrestrial carbon uptake as the implicit ocean carbon uptake is worrying. The temperature and CO₂ responses of the ocean and terrestrial carbon sinks are fundamentally different in nature.

Yes they are, but we would counter that the airborne fraction which drives the behaviour of atmospheric CO₂ concentration change and temperature change from emissions follows historical constraints and produces reasonable projections when integrated forwards with any realistic pathway. FAIR is designed to calculate concentrations, radiative forcing and temperature change from input emissions, which it does well. It is not designed to calculate land and ocean carbon uptake.

The fact that the CO₂ fertilisation and soil carbon residence time respond to temperature and CO₂ independently to ocean carbon uptake in reality, but are lumped together in the FAIR model, also may affect the parameter space results, and the likelihood of remaining under 1.5 °C for any particular pathway. This is because the (implicit) ocean and terrestrial carbon uptake are automatically correlated in the FAIR ensemble, but are independent in reality.

It is true that land and ocean feedback terms are uncorrelated in reality. We again mention that FAIR does not model these processes explicitly, rather seeking to capture the overall evolution of radiative forcing (a function of airborne fraction of CO₂) and its associated impacts on overall temperature, and point to the evidence contained in this response to demonstrate that this is achieved.

References:

Follows, M. J., S. Dutkiewicz, and T. Ito (2006), On the solution of the carbonate system in ocean biogeochemistry models, *Ocean Modelling*, 12,290– 301.

Goodwin et al (2007) Ocean-atmosphere partitioning of anthropogenic carbon dioxide on centennial timescales, *Global Biogeochemical Cycles* 21, doi:10.1029/2006GB002810.

Joos, F., M. Bruno, R. Fink, U. Siegenthaler, T.F. Stocker, C. Le Quere & J.L. Sarmiento, (1996) An efficient and accurate representation of complex oceanic and biospheric models of anthropogenic carbon uptake, *Tellus* 48B, p397-417.

Lenton, T. (2003) Climate Change to the End of the Millennium, *Climatic Change* 76, pp7-29.

Zeebe and Wolf-Gladrow (2001) CO₂ in Seawater: Equilibrium, Kinetics, Isotopes, Elsevier, pp360.

Reviewer #3 (Remarks to the Author):

The revised manuscript by Smith et al. is substantially improved over the original submission. I particularly appreciate the plain language explanations of different types of geophysical inertia, which I think will be quite useful to readers who may have wrong ideas about the climate system and the inevitability of future warming.

However, I think it requires further revisions, as described below.

Thank you for many good suggestions to improve the presentation of results and the perception of trustworthiness to the reader. There are several points which we address in turn.

I am still surprised by the magnitude of temperature decline by the end of the century in many of the paper's simulations, and see that other reviewers were similarly surprised.

We claim that this is due to two factors. The first is a consequence of the non-CO₂ forcing pathway assumed in the phase-out scenarios, dominated by the agricultural assumptions (more specific commentary in response to the question on agriculture, below). The agricultural emissions phase out scenario that we use is more aggressive than RCP2.6, as shown in Figure S11 (formerly S10). To recap, our scenarios assume the green trajectory in that figure, and RCP2.6 is represented by the blue trajectory. Peak temperatures are not much lower in the phase out pathway compared to RCP2.6, but in 2100 the linear phase-out gives end-of-century temperatures that are 0.12°C cooler than RCP2.6. Figure S3 shows the impact of these agricultural assumptions on concentrations of CH₄ and N₂O. The short atmospheric lifetime of methane results in its concentrations returning to pre-industrial levels (<800 ppb) by the end of the century. This is not true in RCP2.6 in which atmospheric concentrations are around 1200 ppb in 2100. The FaIR model simulates this (admittedly MAGICC-derived) trajectory well (Figure 4b in Smith *et al.* (2018)).

The second factor is that values of ECS and TCR that are used to drive our simulations (a) are derived from a joint lognormal (rather than a normal) prior distribution and (b) are constrained based on historical observed temperatures from Cowtan and Way (2014) to effectively form a posterior. Both factors result in prior distributions of ECS and TCR that are typically a little lower than those emergent across the range of CMIP5 models – even though the distribution is itself derived from CMIP5 models – and in particular the distribution used in simulations performed by MAGICC.

I don't find references to discussion papers terribly convincing, and hope that the other reviewers have commented further regarding the suitability and trustworthiness of the FAIR model as used in this study.

The final model version (1.3) is now published in GMD and the references have been updated in the paper. Other than a minor update to the default species-dependent aerosol emission coefficients it remains the same model as was presented in the previous submission. For consistency all model results have been produced using v1.3.4. We explain

in detail in response to reviewer #2 how the airborne fraction of CO₂ is modelled and simulates contemporary earth system models. Non-CO₂ relationships are based on IPCC Fifth Assessment Report relationships, or newer, whereas the two-time-constant response of radiative forcing to surface temperature (incorporating equilibrium climate sensitivity and transient climate response) is as emulated by Geoffroy *et al.* (2013) from a suite of CMIP5 models. We hope that the responses contained in our reply particularly to reviewer #2, plus the comparison to the RCP scenarios in Smith *et al.* (2018) and comparison to earth system models in Jenkins *et al.* (2018) will satisfy the reviewer in the model's ability to simulate a range of future climate pathways.

Given the importance of present-day aerosol forcing to the findings, I think it is important that the paper expand its discussion to include more than just a reference to AR5. Where do estimates of present-day aerosol forcing come from? Do different methods result in systematically different estimates? Can the authors use their results to suggest a rough W/m² threshold of present-day forcing that may determine our ability to avoid 1.5degC of warming in different emissions cases? Not only might that be a useful quantity for aerosol researchers, it would make it clear that the top line results of this paper are dependent on an uncertain parameter.

Thank you for this suggestion. Under the “Model parameter uncertainty analysis” heading the following discussion has been inserted. A new supplementary figure S9 shows the peak temperature versus aerosol ERF for SSP2-2018-ZERO and SSP2-2018-MID.

The prior distribution for year-2011 aerosol forcing is taken from the AR5 “expert judgement” assessed uncertainty range from a combination of CMIP5-era climate models and satellite observations. Climate models tend to produce estimates of aerosol ERF that are more negative than satellite observations, but methodological biases in both procedures prevent a conclusive assessment of which product is more likely to be more representative of the true aerosol ERF (Boucher *et al.* 2013). If present-day (2018) aerosol forcing is towards the less negative end of the uncertainty range, the zero-emissions commitment in SSP2-2018-ZERO is small; a lower bound of -1 W m^{-2} produces a peak warming that remains under 1.5°C across all ensemble members (Supplementary Fig. 9). For some small negative values of present-day aerosol ERF, temperatures start to fall immediately after the zero-emissions commitment suggesting that the decline in other SLCFs, mostly tropospheric ozone and methane, leads to a cooling that more than offsets a warming from aerosol removal under this scenario. For the SSP2-2018-MID infrastructure commitment there is a larger spread in peak temperature owing to the increased importance of other factors aside from the present-day aerosol forcing. These include climate sensitivity and the thermal response of the ocean, both of which have an extra few decades to act on the temperature evolution compared to SSP2-2018-ZERO. However, peak temperature remains below 1.5°C in the SSP2-2018-MID ensemble when present-day aerosol ERF is less negative than -0.6 W m^{-2} .

The phase-out scenarios are better than in the prior version, but the main text does a poor job explaining that these scenarios are not bottom-up (i.e. unit-level) projections, but rather rough estimates of emissions reductions possible based on prior work. I think it is important to clarify this. A good way to do this would be to report the mean

annual mitigation rate inferred under Fast, Mid, and Slow phase-out scenarios. I believe this is essentially what the authors are calculating from previous studies and applying in a stylized way to estimate the future rate of decline of present-day emissions. For context, the actual rate could also be reported for the past decade.

Thank you for this suggestion. A paragraph has been added in the Scenarios and Model section:

Our phase-out scenarios are not bottom-up sectoral emissions projections or derived from detailed modelling. Instead, they assess the potential speed of emissions phase-out of each sector based on earlier literature and asset lifetimes. The scenarios assume mean rates of decarbonisation of 0.30, 0.23 and 0.18 GtC yr⁻¹ for the FAST, MID and SLOW pathways over the first 30, 40 or 50 years respectively (Supplementary Table 1). In comparison, the mean rate of emissions increase over 2006-2016 was 0.15 GtC yr⁻¹ (Le Quéré *et al.* 2018).

Citations to papers by Mark Jacobson are carrying a lot of weight in lines 52-53, where they support the proposition that it “may be technically and economically possible [to transition to a zero carbon energy system].” This is highly controversial, and a diverse group of energy modelers have argued that Jacobson’s work demonstrates no such thing (Clack et al., 2017). Regardless of which side of the debate the authors are on, it seems important to more explicitly acknowledge the controversy.

The reviewer raises a good point. We used Jacobson’s references to reflect an opinion that essentially justified investigating these pathways. The Clack reference, along with the one from Ted Trainer, provides the opposing perspective. The sentences have been updated:

Transitioning to a zero carbon energy system within 40 years will be politically and societally challenging to say the least, and opinions are divided on whether this may be technically and economically possible (Jacobson and Delucchi 2011; Bruckner *et al.* 2014; Delucchi and Jacobson 2011; Trainer 2010; Clack *et al.* 2017). We do not seek to assess the practical feasibility of this transition, but merely to report on the consequences in the context of the Paris Agreement.

Regarding the phase-out of agricultural emissions over the remainder of the century, this is in line with the baseline in the SSPs. However, it is not clear to me if such a phase-out implies large-scale deployment of negative emissions technologies, which many analysts might deem fantastical. To the extent their conclusions may depend on such fantasy, the authors should be clear and discuss. There is a growing literature on this subject, including several important pieces in a recent volume of ERL.

The treatment of agricultural emissions phase out requires a more nuanced decision, as they clearly don’t relate to fossil fuel burning or have a clearly definable “asset lifetime” other than for livestock. Unlike many of the primarily CO₂-producing sectors there is no suitable net-zero substitute that could sustain global food production.

There are no negative emissions technologies assumed for agriculture, as methane and nitrous oxide emissions dominate. Any CO₂ land-use emissions from agriculture are combined with land-use CO₂ from other sectors to form a total AFOLU contribution which is set to zero in 2019.

In the Methods section, the discussion on agriculture has been expanded.

Finally, I encourage the authors to re-read their text carefully, as there remain worrisome errors of language. For example, this first sentence of the “Summary and outlook” section indicates that “limiting warming to 1.5degC...occurs in only 10% of ensemble members...” This is the opposite of what the authors conclude.

We agree that this is unclear and thank the reviewer for pointing it out. It has been changed thus:

In this paper we show that limiting warming to 1.5°C is not yet geophysically impossible. Exceeding 1.5°C occurs in only 9% of ensemble members under a zero emissions commitment if emissions cease at the end of 2018.

References used in this response:

- Boucher, O., D. Randall, P. Artaxo, C. Bretherton, G. Feingold, P. Forster, V. M. Kerminen, Y. Kondo, H. Liao, U. Lohmann, P. Rasch, S. K. Satheesh, S. Sherwood, B. Stevens and X. Y. Zhang. 2013. Clouds and Aerosols. *In*: T. F. STOCKER, D. QIN, G. K. PLATTNER, M. TIGNOR, S. K. ALLEN, J. BOSCHUNG, A. NAUELS, Y. XIA, V. BEX and P. M. MIDGLEY, eds. *Climate Change 2013: The Physical Science Basis. Contribution of Working Group I to the Fifth Assessment Report of the Intergovernmental Panel on Climate Change*. Cambridge, United Kingdom and New York, NY, USA: Cambridge University Press, pp.571-658.
- Bruckner, T., I. A. Bashmakov, Y. Mulugetta, H. Chum, A. De La Vega Navarro, J. Edmonds, A. Faaij, B. Fungtammasan, A. Garg, E. Hertwich, D. Honnery, D. Infield, M. Kainuma, S. Khennas, S. Kim, H. B. Nimir, K. Riahi, N. Strachan, R. Wiser and X. Zhang. 2014. Energy Systems. *In*: O. EDENHOFER, R. PICHS-MADRUGA, Y. SOKONA, E. FARAHANI, S. KADNER, K. SEYBOTH, A. ADLER, I. BAUM, S. BRUNNER, P. EICKEMEIER, B. KRIEMANN, J. SAVOLAINEN, S. SCHLÖMER, C. VON STECHOW, T. ZWICKEL and J. C. MINX, eds. *Climate Change 2014: Mitigation of Climate Change. Contribution of Working Group III to the Fifth Assessment Report of the Intergovernmental Panel on Climate Change*. Cambridge, United Kingdom and New York, NY, USA: Cambridge University Press.
- Clack, C. T. M., S. A. Qvist, J. Apt, M. Bazilian, A. R. Brandt, K. Caldeira, S. J. Davis, V. Diakov, M. A. Handschy, P. D. H. Hines, P. Jaramillo, D. M. Kammen, J. C. S. Long, M. G. Morgan, A. Reed, V. Sivaram, J. Sweeney, G. R. Tynan, D. G. Victor, J. P. Weyant and J. F. Whitacre. 2017. Evaluation of a proposal for reliable low-cost grid power with 100% wind, water, and solar. *Proceedings of the National Academy of Sciences*, **114**(26), pp.6722-6727.
- Cowtan, K. and R. G. Way. 2014. Coverage bias in the HadCRUT4 temperature series and its impact on recent temperature trends. *Q. J. Roy. Meteor. Soc.*, **140**(683), pp.1935-1944.

- Delucchi, M. A. and M. Z. Jacobson. 2011. Providing all global energy with wind, water, and solar power, Part II: Reliability, system and transmission costs, and policies. *Energy Policy*, **39**, pp.1170-1190.
- Friedlingstein, P., P. Cox, R. Betts, L. Bopp, W. Von Bloh, V. Brovkin, P. Cadule, S. Doney, M. Eby, I. Fung, G. Bala, J. John, C. Jones, F. Joos, T. Kato, M. Kawamiya, W. Knorr, K. Lindsay, H. D. Matthews, T. Raddatz, P. Rayner, C. Reick, E. Roeckner, K. G. Schnitzler, R. Schnur, K. Strassmann, A. J. Weaver, C. Yoshikawa and N. Zeng. 2006. Climate–Carbon Cycle Feedback Analysis: Results from the C4MIP Model Intercomparison. *J. Climate*, **19**(14), pp.3337-3353.
- Frölicher, T. L., M. Winton and J. L. Sarmiento. 2014. Continued global warming after CO₂ emissions stoppage. *Nature Climate Change*, **4**(1), pp.40-44.
- Geoffroy, O., D. Saint-Martin, D. J. L. Olivié, A. Voldoire, G. Bellon and S. Tytéca. 2013. Transient Climate Response in a Two-Layer Energy-Balance Model. Part I: Analytical Solution and Parameter Calibration Using CMIP5 AOGCM Experiments. *J. Climate*, **26**(6), pp.1841-1857.
- Goodwin, P., R. G. Williams, M. J. Follows and S. Dutkiewicz. 2007. Ocean-atmosphere partitioning of anthropogenic carbon dioxide on centennial timescales. *Global Biogeochemical Cycles*, **21**(1).
- Jacobson, M. Z. and M. A. Delucchi. 2011. Providing all global energy with wind, water, and solar power, Part I: Technologies, energy resources, quantities and areas of infrastructure, and materials. *Energy Policy*, **39**, pp.1154-1169.
- Jenkins, S., R. J. Millar, N. Leach and M. R. Allen. 2018. Framing Climate Goals in Terms of Cumulative CO₂-Forcing-Equivalent Emissions. *Geophysical Research Letters*, **45**(6), pp.2795-2804.
- Joos, F., R. Roth, J. S. Fuglestedt, G. P. Peters, I. G. Enting, W. Von Bloh, V. Brovkin, E. J. Burke, M. Eby, N. R. Edwards, T. Friedrich, T. L. Frölicher, P. R. Halloran, P. B. Holden, C. Jones, T. Kleinen, F. T. Mackenzie, K. Matsumoto, M. Meinshausen, G. K. Plattner, A. Reisinger, J. Segschneider, G. Shaffer, M. Steinacher, K. Strassmann, K. Tanaka, A. Timmermann and A. J. Weaver. 2013. Carbon dioxide and climate impulse response functions for the computation of greenhouse gas metrics: a multi-model analysis. *Atmos. Chem. Phys.*, **13**(5), pp.2793-2825.
- Katavouta, A., R. G. Williams, P. Goodwin and V. M. Roussenov. 2018. Reconciling Atmospheric and Oceanic Views of the Transient Climate Response to Emissions. *Geophysical Research Letters*, **45**.
- Le Quéré, C., R. M. Andrew, P. Friedlingstein, S. Sitch, J. Pongratz, A. C. Manning, J. I. Korsbakken, G. P. Peters, J. G. Canadell and R. B. Jackson. 2018. Global Carbon Budget 2017. *Earth System Science Data*, **10**(1), pp.405-448.
- Leach, N. J., R. J. Millar, K. Haustein, S. Jenkins, E. Graham and M. R. Allen. 2018. Current level and rate of warming determine emissions budgets under ambitious mitigation. *Nature Geoscience*, **11**(8), pp.574-579.
- Mauritsen, T. and R. Pincus. 2017. Committed warming inferred from observations. *Nature Climate Change*, **7**(9), pp.652-655.
- Meinshausen, M., N. Meinshausen, W. Hare, S. C. B. Raper, K. Frieler, R. Knutti, D. J. Frame and M. R. Allen. 2009. Greenhouse-gas emission targets for limiting global warming to 2 C. *Nature*, **458**(7242), pp.1158-1162.
- Meinshausen, M., S. J. Smith, K. V. Calvin, J. S. Daniel, M. L. T. Kainuma, J. F. Lamarque, K. Matsumoto, S. A. Montzka, S. C. B. Raper, K. Riahi, A. M. Thomson, G. J. M. Velders and D. Van Vuuren. 2011. The RCP Greenhouse Gas Concentrations and their Extension from 1765 to 2300. *Climatic Change*.
- Millar, R. J., Z. R. Nicholls, P. Friedlingstein and M. R. Allen. 2017. A modified impulse-response representation of the global near-surface air temperature and atmospheric concentration response to carbon dioxide emissions. *Atmos. Chem. Phys.*, **17**, pp.7213-7228.
- Plattner, G. K., R. Knutti, F. Joos, T. F. Stocker, W. Von Bloh, V. Brovkin, D. Cameron, E. Driesschaert, S. Dutkiewicz, M. Eby, N. R. Edwards, T. Fichet, J. C. Hargreaves, C.

- D. Jones, M. F. Loutre, H. D. Matthews, A. Mouchet, S. A. Müller, S. Nawrath, A. Price, A. Sokolov, K. M. Strassmann and A. J. Weaver. 2008. Long-Term Climate Commitments Projected with Climate–Carbon Cycle Models. *Journal of Climate*, **21**(12), pp.2721-2751.
- Revelle, R. and H. E. Suess. 1957. Carbon Dioxide Exchange Between Atmosphere and Ocean and the Question of an Increase of Atmospheric CO₂ during the Past Decades. *Tellus*, **9**(1), pp.18-27.
- Smith, C. J., P. M. Forster, M. Allen, N. Leach, R. J. Millar, G. A. Passerello and L. A. Regayre. 2018. FAIR v1.3: A simple emissions-based impulse response and carbon cycle model. *Geoscientific Model Development*, **11**(6), pp.2273-2297.
- Trainer, T. 2010. Can renewables etc. solve the greenhouse problem? The negative case. *Energy Policy*, **38**(8), pp.4107-4114.

Reviewer #2 (Remarks to the Author):

The authors have taken the time to demonstrate that the FaIR emulator is applicable for the experiments discussed in this manuscript.

The latest version of the manuscript reads well, with findings that are well supported by the experiments conducted and with a methods section that accurately describes the experiments themselves.

The findings within the manuscript are significant, and will be of interest to a wide audience, and so are worthy of publication within Nature Communications.

I support the publication of this manuscript in its present form.

Reviewer #3 (Remarks to the Author):

I am satisfied by the edits the authors made in response to my comments.

I have also read the comments of reviewer 1 and the authors' responses thereto. I am somewhat persuaded of the FaIR model's suitability based on the comparison plots provided by the authors, but it would be important to me that the Reviewer was persuaded, as the conclusions of the paper are almost entirely based on the FaIR model, their claims are strong, and the caveats they have added in response to my comments are not especially prominent.